# Improved rain event detection in Commercial Microwave Link time series via combination with MSG SEVIRI data

Maximilian Graf[1,*], Andreas Wagner[1,*], Julius Polz[2,*], Llorenç Lliso[3], José Alberto Lahuerta[3], Harald Kunstmann[1,2], and Christian Chwala[2]

[1]University of Augsburg, Institute of Geography (IGUA), Alter Postweg 118, 86159 Augsburg, Germany
[2]Karlsruhe Institute of Technology, Campus Alpin (IMK-IFU), Kreuzeckbahnstraße 19, 82467 Garmisch-Partenkirchen, Germany
[3]Agencia Estatal De Meteorología (AEMET Spain)
[*]These authors contributed equally to this work.

**Correspondence:** Christian Chwala (christian.chwala@kit.edu)

**Abstract.** The most reliable areal precipitation estimation is usually generated via combinations of different measurements. Path-averaged rain rates can be derived from Commercial Microwave Links (CML), where attenuation of the emitted radiation is strongly related to rainfall rate. CMLs can be combined with data from other rainfall measurements or used individually. They are available almost worldwide and often represent the only opportunity for ground-based measurement in data-scarce regions. However, deriving rainfall estimates from CML data requires extensive data processing. The separation of the attenuation time series into rainy and dry periods (rain event detection) is the most important step in this processing and has a high impact on the resulting rainfall estimates. In this study, we investigate the suitability of Meteosat Second Generation Spinning Enhanced Visible and InfraRed Imager (MSG SEVIRI) satellite data as an auxiliary-data-based (ADB) rain event detection method. We compare this method with two time-series-based (TSB) rain event detection methods. We used data from 3748 CMLs in Germany for four months in the summer of 2021 and data from the two SEVIRI-derived products *PC* and *PC-Ph*. We analyzed all rain event detection methods for different rainfall intensities, differences between day and night, as well as their influence on the performance of rainfall estimates from individual CMLs. The radar product RADKLIM-YW was used for validation. The results showed that both SEVIRI products are promising candidates for ADB rainfall detection, yielding only slightly worse results than the TSB methods with the main advantage, that the ADB method does not rely on extensive validation for different CML datasets. The main uncertainty of all methods was found for light rain. Slightly better results were obtained during the day than at night due to the reduced availability of SEVIRI channels at night. In general, the ADB methods led to improvements for CMLs performing comparatively weakly using TSB methods. Based on these results, combinations of ADB and TSB methods were developed by emphasizing their specific advantages. Compared to basic and advanced TSB methods, these combinations improved the Matthews Correlation Coefficient of the rain event detection from 0.49 (0.51 resp.) to 0.59 during the day and from 0.41 (0.50 resp.) to 0.55 during the night. Additionally, these combinations increased the number of true positive classifications, especially for light rainfall compared to the TSB methods, and reduced the number of false negatives while only leading to a slight increase in false positive classifications. Our results show that utilizing MSG SEVIRI data in CML data processing significantly increases the quality of the rain event detection step, in particular for CMLs which

are challenging to process with TSB methods. While the improvement is useful even for applications in Germany, we see the main potential of using ADB methods in data-scarce regions like West Africa where extensive validation is not possible.

## 1 Introduction

Rainfall is the most important variable for hydrology and water management. It is characterized by high variability in space and time, especially in the case of convective rain events. The quality of hydrological modeling results depends heavily on high-resolution and reliable areal rainfall data (Fu et al., 2011; Bruni et al., 2015; Rafieeinasab et al., 2015; Cristiano et al., 2017).

There are a variety of rainfall measurement methods that serve as a basis for the derivation of rainfall fields, each with specific advantages but also drawbacks. Rain gauges can provide point measurements of precipitation with high accuracy, but they are prone to errors due to wind and evaporation (Sevruk, 2006) and primarily lack spatial representativeness (Pollock et al., 2018). Satellite data provide areal precipitation estimates almost worldwide with a spatial resolution in the order of several kilometers. But they either suffer from a poor temporal resolution like the GPM core satellite that has a revisit time of approximately 1 day in the tropics or from heterogeneous data quality and delayed availability like merged satellite products like IMERG (Hou et al., 2014). Additionally, complex retrieval and calibration algorithms have to be applied which cause additional uncertainties (Maggioni et al., 2016).

Weather radars derive areal precipitation estimates with a high resolution of 5 minutes and 1 km (Atlas, 1990; Bartels et al., 2004; Winterrath et al., 2012). However, the calculation of rain rates from radar reflectivity is non-trivial (Uijlenhoet et al., 2003; Steiner et al., 2004) and false echoes, clutter, and other measurement effects cause further problems (Villarini and Krajewski, 2010; Wagner et al., 2012; Wagner, 2018). Schleiss et al. (2020) showed that radar data tends to underestimate particularly heavy rainfall in Scandinavian countries. Nevertheless, gauge-adjusted radar products are considered to be one of the best possible data basis for spatial rainfall estimates, because they leverage the advantages from individual measurement devices (Bartels et al., 2004; Winterrath et al., 2012).

The opportunistic sensing of rainfall with Commercial Microwave Links (CML) was first demonstrated in Israel (Messer et al., 2006) and the Netherlands (Leijnse et al., 2007). In recent years CML rainfall estimation has become available on country-wide scales (Overeem et al., 2016; Graf et al., 2020). Rainfall attenuates the microwave radiation between two antennas of a CML. The relationship between attenuation and rainfall is close to linear for signals between 10 and 40 GHz (Atlas and Ulbrich, 1977). CMLs have already proven their potential as stand-alone rainfall sensors in multiple regions of the world (Overeem et al., 2013; Rios Gaona et al., 2015; Overeem et al., 2016; D'Amico et al., 2016; Graf et al., 2020; Roversi et al., 2020; van de Beek et al., 2020; Djibo et al., 2023).

Additionally, the path averaged rainfall information from CMLs can complement conventional measurement methods (Liberman et al., 2014; Haese et al., 2017; Graf et al., 2021). Kumah et al. (2022) for instance derived rain intensities from MSG satellite data by a random forest algorithm trained with CML rainfall estimates. For regions with sparse observation networks like most parts of Africa, where radar data is missing and even station data is only sparsely available, CMLs can deliver addi-

tional ground-based rainfall estimates (Djibo et al., 2023). Thus, the use of CML data is a good opportunity to reduce the gap in the global availability of climate information as recently emphasized by UNFCCC (2022).

The detection of rain events in CML attenuation time series is an important processing step for several reasons. First, it defines the rainy periods for which a baseline has to be defined, typically from preceding dry time steps. Second, it filters fluctuations that are not caused by rain, but by other disturbances e.g. from refraction, multi-path propagation, or mast sway (c.f. Chwala and Kunstmann (2019) for a detailed list). For an overview of available rain event detection methods, we divide them into two categories, time series-based (TSB) methods and methods based on auxiliary data of rainfall patterns (ADB). ADB methods that are based on globally available data like the satellite data presented in this study are especially promising for CML processing in regions with otherwise sparse rainfall information.

Examples of TSB methods are the simple threshold models (Leijnse et al., 2008), an approach using the rolling standard deviation (Schleiss and Berne, 2010), Markov switching models (Wang et al., 2012) or short-term Fourier Transform approaches (Chwala et al., 2012). Messer and Sendik (2015) provide a detailed description of these approaches. Machine Learning approaches to distinguish between wet and dry time steps emerged in recent years, usually outperforming the previous methods (Habi and Messer, 2018; Polz et al., 2020; Song et al., 2020). The "nearby-link" approach (Overeem et al., 2016) is a hybrid TSB and ADB method because it compares CML attenuation time series of neighboring CMLs. Similar to CMLs, satellite microwave links (SMLs) can be exploited to derive rainfall estimates. SML processing also includes rain event detection and several methods are available (Giannetti et al., 2019; Giro et al., 2022).

Data sources in ADB methods can be weather radar (Overeem et al., 2011) or satellite data (van het Schip et al., 2017; Kumah et al., 2021). Regarding satellite data, geostationary satellites such as MSG SEVIRI offer a temporal resolution of 15 minutes at 4x6 km spatial resolution in mid-latitudes. This data is also used as areal precipitation (Roebeling et al., 2008; Roebeling and Holleman, 2009), although the derivation of precipitation from VIS and IR channels is often uncertain. According to NWC SAF (Lahuerta García, 2021) even a distinction between light, moderate, and heavy precipitation is difficult. That is the reason why they determine the probability of precipitation in their post-processed SEVIRI products.

van het Schip et al. (2017) analyzed applying post-processed SEVIRI products as a wet-dry indicator in the Netherlands. They evaluated the resulting rainfall maps for 12 days and found improvements compared to not separating the time series into wet and dry periods but a decreased performance compared to radar-based rain event detection. However, they did not compare the methods for individual CMLs, different rainfall intensities, or day and night periods for which the satellite products use different channels and methodologies. They also did not combine their ADB method with a TSB method.

Kumah et al. (2021) obtained improved rain intensities for convective rain events when applying MSG SEVIRI data for rain event detection of CML data. However, their results are based only on one CML in Kenya for daytime and rain intensities above 0.5 mmh$^{-1}$. The limitation of both studies regarding the analyzed period, the number of CMLs, intensity classes, and day and night differences limit the transferability of their results. None of the presented rain event detection methods can provide a high-quality classification for CML datasets with varying characteristics (e.g. sparse or dense network, various temporal

**Table 1.** Overview of rainfall sensors and products.

| Sensor | Product | Spatial resolution (longitude x latitude) | Temporal resolution | Data points (including missing values) |
|---|---|---|---|---|
| C-Band weather radar | RADKLIM-YW | 1km x 1km | 5 minutes | 30,808,350,000 |
| MSG SEVIRI | *PC* & *PC-Ph* | 3.4-5.1km x 4.0-6.4km | 15 minutes | 1,330,560,000 |
| CML | - | 3748 CML paths | 1 min | 765,649,270 |

resolutions, different frequency ranges, etc.). CMLs with frequencies below 10 GHz which are commonly deployed in sparse CML networks in rural Sub-Saharan Africa still pose great challenges for all available rain event detection methods. Yet, these regions are often associated with a high potential for CML as the sole source of rainfall estimates.

In this study, the precipitation products *PC* and *PC-Ph*, which are computed by NWC SAF using data from the SEVIRI radiometer onboard the geostationary satellite METEOSAT, are used to classify CML attenuation time series in rainy and dry periods. In addition to comparing ADB (based on *PC* and *PC-Ph*) and TSB (CML time series processing) methods, this study presents a novel way of combining TSB and ADB rain event detection approaches to improve rain event detection. To analyze the applicability of such new rain event detection methods, data sets of high data quality are necessary. The present analysis

is based on country-wide CML data from 4 months in the summer of 2021 in Germany. As reference a high-resolution gauge-adjusted radar product is used.

The research questions of this investigation are: (1) are *PC* and *PC-Ph* products suitable as wet-dry indicators for CML data? (2) Do the results vary with rain intensity? (3) Are there noticeable differences between day and night? (4) Can a combination

of TSB and ADB rain event detection methods outperform TSB-only and ADB-only methods?

## 2 Data

This study is based on CML, weather radar, and SEVIRI data in Germany covering the period from the 30th of April 2021 to the 10th of September 2021. Tab. 1 summarizes the properties of these products.

### 2.1 CML data

The CML dataset consists of a subset of CMLs operated by Ericsson in Germany. For the analyzed period, 3748 CMLs were available for more than 30% of the time and thus considered in this study. The path length of the CMLs varies between 0.2 km and about 30 km and the frequencies range from 10 to 40 GHz with shorter CMLs using higher frequencies (see Fig.1). The transmitted signal level (TSL) with a power resolution of 1 dB and the received signal level (RSL) with a power resolution of 0.3 dB are instantaneously measured every minute using a custom real-time CML data acquisition system (Chwala et al.,

2016). The total loss (TL) is the difference between TSL and RSL. Each CML consists of two sublinks for two-way data transmission. The processing of the data is described in Section 3.2.

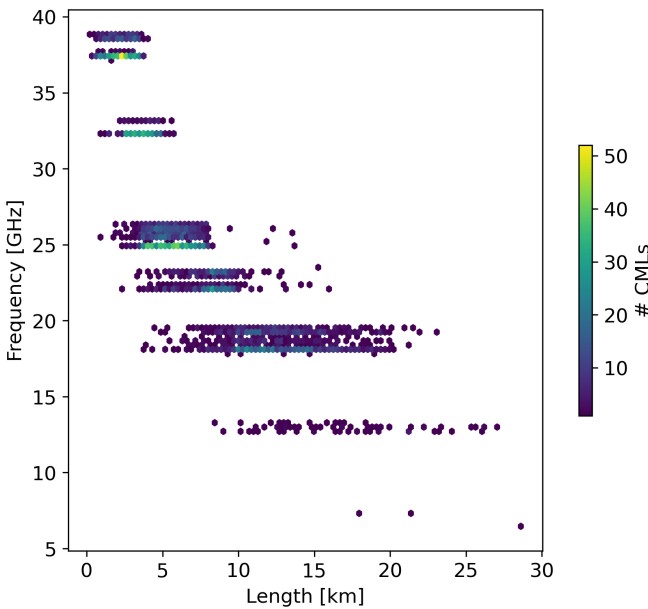

**Figure 1.** Scatter density plot showing the distribution of length and frequency of the 3748 CMLs in Germany that are used in this study.

## 2.2 Weather radar data

*RADKLIM-YW* is a gauge-adjusted, climatologically-corrected radar product of the German Meteorological Service (DWD). The hourly adjustment with data from automatic rainfall stations is identical to the one of the *RADOLAN-RW* product (Bartels et al., 2004; Winterrath et al., 2012). Daily sums are adjusted with daily measurements of manual rainfall stations and climatology-based corrections of spokes and range-dependencies are carried out (Winterrath et al., 2018). *RADKLIM-YW* data has a temporal resolution of 5 minutes and a spatial resolution of 1 km covering Germany with 1100 times 900 grid cells. According to Kreklow et al. (2020), the range-dependent effects, which are particularly strong in winter, are reduced. This improves rainfall patterns and better represents orography compared to *RADOLAN-RW*. At the same time, however, a slight underestimation of the total rainfall was determined.

*RADKLIM-YW* is used to validate the binary classifications of rain event detection methods and the rainfall estimates derived from CMLs using these methods. To compare the gridded *RADKLIM-YW* with the CMLs we averaged the grid for each individual CML path weighted by the length of intersecting path segments in each pixel. *RADKLIM-YW* was aggregated to a resolution of 15 minutes. For the usage as a binary wet-dry reference, we used a rainfall intensity threshold of 0.1 mm/h at the 15-minute resolution. All values below 0.1 mm/h were considered dry.

To answer the research question (2), whether the performance of rain event detection methods shows a rain rate dependency, we define intensity classes and group the 15-minute radar rainfall intensities. The classes light (0.1 to 2.5 mmh$^{-1}$), moderate (2.5 to 10.0 mmh$^{-1}$), and heavy (more than 10.0 mmh$^{-1}$) rainfall are based on the (DWD, 2023) classification. Light rainfall

is further subdivided into the classes light1 (0.1 to 1.0 mmh$^{-1}$) and light2 (1.0 to 2.5 mmh$^{-1}$). Values below 0.1 mmh$^{-1}$ are considered dry. The resulting classes are shown in Fig.7.

## 2.3 SEVIRI data

The Spinning Enhanced Visible and Infrared Imager (SEVIRI) radiometer from the geostationary satellite METEOSAT provides image data in two visible (VIS), one high-resolution channel (HRVIS), and nine Infrared (IR) channels including one near infrared channel (NIR). The channels range from 0.5 to 14.4 micrometers with a resolution of 3 km at the sub-satellite point. The high-resolution channel is not used for our purposes. Every 15 minutes a calibrated image of the full earth disc (lon: -79° to 79°, lat: -81° to 81°) is available (Schmid, 2000). No space-borne radar or radiometer for precipitation measurement is on board, such as for GPM (Hou et al., 2014). Precipitation products are derived based on two approaches: either by regression of different channels and adjustment to precipitation measurements or, more sophisticatedly, by deriving microphysical parameters (Roebeling et al., 2008; Hernanz et al., 2019). For the calculation of microphysical parameters, the 0.6 um channel (VIS) and the 1.6 um channel (NIR) are mandatory (Roebeling et al., 2008).

The first product *PC-Ph* is derived from the microphysical parameters Effective Radius (Reff) and Cloud Optical Thickness (COT) during daytime. At night, when visible channels are missing, it is calculated by a regression of infrared and water vapor channels. The second product *PC* relies on different regression functions of available SEVIRI channels, including VIS, IR, and WV channels at daytime and only IR and WV channels at night. The definition of day and night time is derived from the variable *pccond* that is provided for both *PC* and *PC-Ph*. Both products are provided by NWC SAF. Recent data is freely available, but long-term records must be requested individually.

We chose both products for this study because they provide the probability of precipitation in percent: *PC* in increments of 10 and *PC-Ph* in increments of 1 percent. Compared to a pure precipitation product, the precipitation probability products enabled us to consistently alter the classification threshold similar to how it can be done for certain TSB methods. Example time series of the precipitation probability are shown in Fig 2 b) and c). More detailed descriptions and the evaluation of the two products are available from (Thoss, 2014; Hernanz et al., 2019) and (Lahuerta García, 2019). Similar to *RADKLIM-YW*, we derived *PC* and *PC-Ph* values along the CML paths. The validation of *PC* or *PC-Ph* wet and dry labels is based on the path averages of *RADKLIM-YW*.

## 3 Methods

### 3.1 Rain event detection

#### 3.1.1 Individual methods for rain event detection

We used two existing time series-based (TSB) methods as a baseline for rain event detection. The first one is based on the rolling standard deviation of total loss (TL). Time steps for which the standard deviation of a 60-minute rolling window exceeds a certain threshold are considered wet. This approach was originally suggested by Schleiss and Berne (2010) who

used a fixed threshold. Later, Graf et al. (2020) determined the threshold based on the 80th percentile of the 60-minute rolling standard deviation of TL multiplied by a scaling factor of 1.12 that adapted the threshold to the general amount of noisiness of each CML. This method will be called (*RS*). The second method is a machine learning approach based on a convolutional neural network that was trained to classify TL time series into rainy and dry time steps from Polz et al. (2020). This model provides a continuous probability between 0 and 1 that describes the likeliness that a time step is rainy. Therefore, the choice

of threshold that divides the probability values into rainy and dry time steps determines whether classification is more liberal or conservative. We adopted the classification threshold of 0.82 which was found to be optimal by Polz et al. (2020). This method will be called *CNN*. Fig. 2a) shows an example of the *CNN* probability and the three thresholds used later in the combination of rain event detection methods. Both TSB approaches were compared in Polz et al. (2020) based on hourly data with significantly better performance of *CNN* compared to *RS*. Identically to Polz et al. (2020), we computed *RS* and *CNN* based on 1-minute

TL data.

We used *PC* and *PC-Ph* products from SEVIRI data as ADB rain event detection by applying a threshold on their precipitation probabilities. We used the thresholds 30 %, 20 %, 10 % and 0.1 %. The last threshold represents probabilities greater than 0 %. The abbreviations for the thresholds are e.g., *P30* or *P01*, and the abbreviations for the specific data sets of *PC* and *PC-Ph* are e.g., *PC10* or *PC-Ph10*. We forward-filled the 15-minute classification to a 1-minute resolution in the CML processing

described in Sec. 3.2. This temporal resolution is necessary for the two TSB methods and other CML processing methods such as WAA compensation and baseline estimation as they were developed and tested for this resolution (Graf et al., 2020).

### 3.1.2 Combinations of rain event detection methods

The main goal of this study is to improve rain event detection compared to already available methods by combining TSB and ADB methods. Using TSB methods, detecting light rain events is more difficult than detecting strong rain events because it is

harder to differentiate smaller attenuation from fluctuations induced by other factors than rain. ADB methods do not use TL for rain event detection, overcoming this issue, but in the case of SEVIRI, uncertainties are introduced by the indirect measurement principle and the difficulties of separating light rain from no rain. We therefore propose to combine TSB and ADB methods to exploit their advantages. We use different probability thresholds for CNN and the two SEVIRI products to derive rain event detection variants with either high confidence in the correct classification of rainy time steps or high confidence regarding dry

time steps. The used thresholds are shown in Fig. 2. *CNN10* and *PC01* are liberal variants in the sense that they classify rainy time steps already for low probabilities, potentially introducing many FPs. Hence, time steps that are predicted to be dry have a lower chance of being FN. Vice versa, *CNN94* and *PC30* are considered conservative variants because they only classify time steps with a very high probability as rainy. This leads to a low number of FP while introducing more FN.

Our procedure of combining the rain event detection variants consists of five individual steps presented in Fig. 3 and algorithm

1 shown in Fig. A1. We either combined *CNN* and *PC* or *CNN* and *PC-Ph*. In the following the combination of *CNN* and *PC* is explained. In step 1, we choose a method with an intermediate threshold as a starting point which is either *CNN82*, *PC01*, or *PC10*. Step 2 uses the dry time steps from the liberal variant *PC10* (even if it was used as the starting point) that has a high confidence for dry time steps to replace rainy time steps from Step 1. Steps 3 and 4 use time steps with high confidence to be

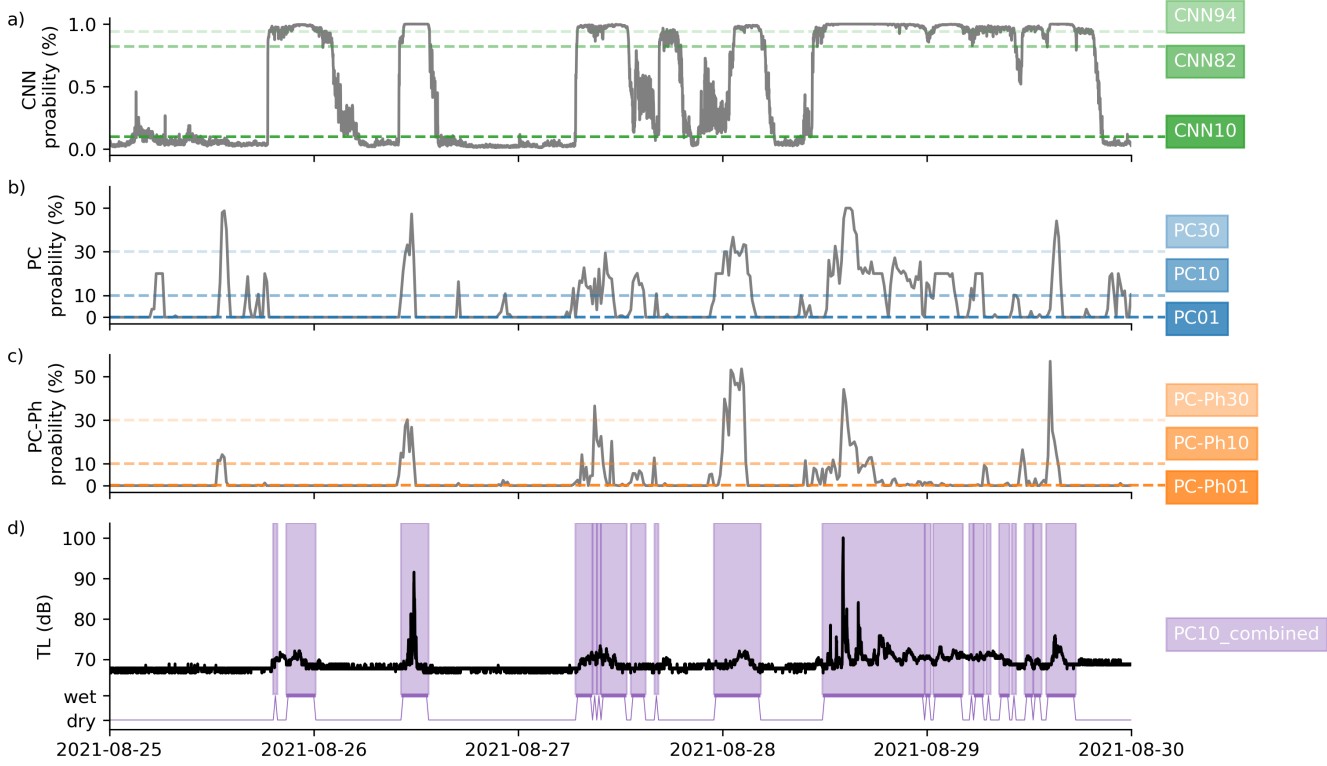

**Figure 2.** Example time series of five days showing the probability for rainy time steps from a) *CNN*, b) *PC*, and c) *PC-Ph* for the TL time series in d). The dashed lines in a), b), and c) represent the thresholds that separate the probabilities into rainy and dry time steps. Higher thresholds (*CNN94*, *PC30*, *PC-Ph30*) lead to a conservative rain event detection, classifying only time steps as rainy which have a high probability of being rainy. Vice versa, low thresholds (*CNN10*, *PC10*, *PC-Ph01* ) lead to a liberal rain event detection, only time steps that are very likely dry are dry. Additionally, the combination of *PC10* with *CNN* and *PC* variants as rain event detection methods is shown in d).

rainy from *CNN94* and *PC30* to replace time steps classified as dry after Step 2. In step 5, dry time steps from *CNN10* are used to replace previously rainy time steps. The combination of *CNN* and *PC-Ph* is identical using *PC-Ph* with the same thresholds instead of *PC*. The results of six combinations are shown in Fig. 5. We named the combinations based on their initial product used for Step 1 as the other steps were identical for each combination e.g. *PC10-combined*

In total, we will evaluate two TSB methods (*RS* and *CNN*, eight ADB methods (derived from the two SEVIRI products *PC* and *PC-Ph* with four thresholds respectively), and six combinations of TSB and ADB methods.

## 3.2   CML data processing

Deriving rainfall estimates from CMLs is a delicate matter (Uijlenhoet et al., 2018). Various research groups developed individual CML processing methods that depend on e.g. the sampling strategy of data. The most important aspects of data processing are briefly outlined hereafter while a more detailed description of the processing steps can be found in Graf et al. (2020). We

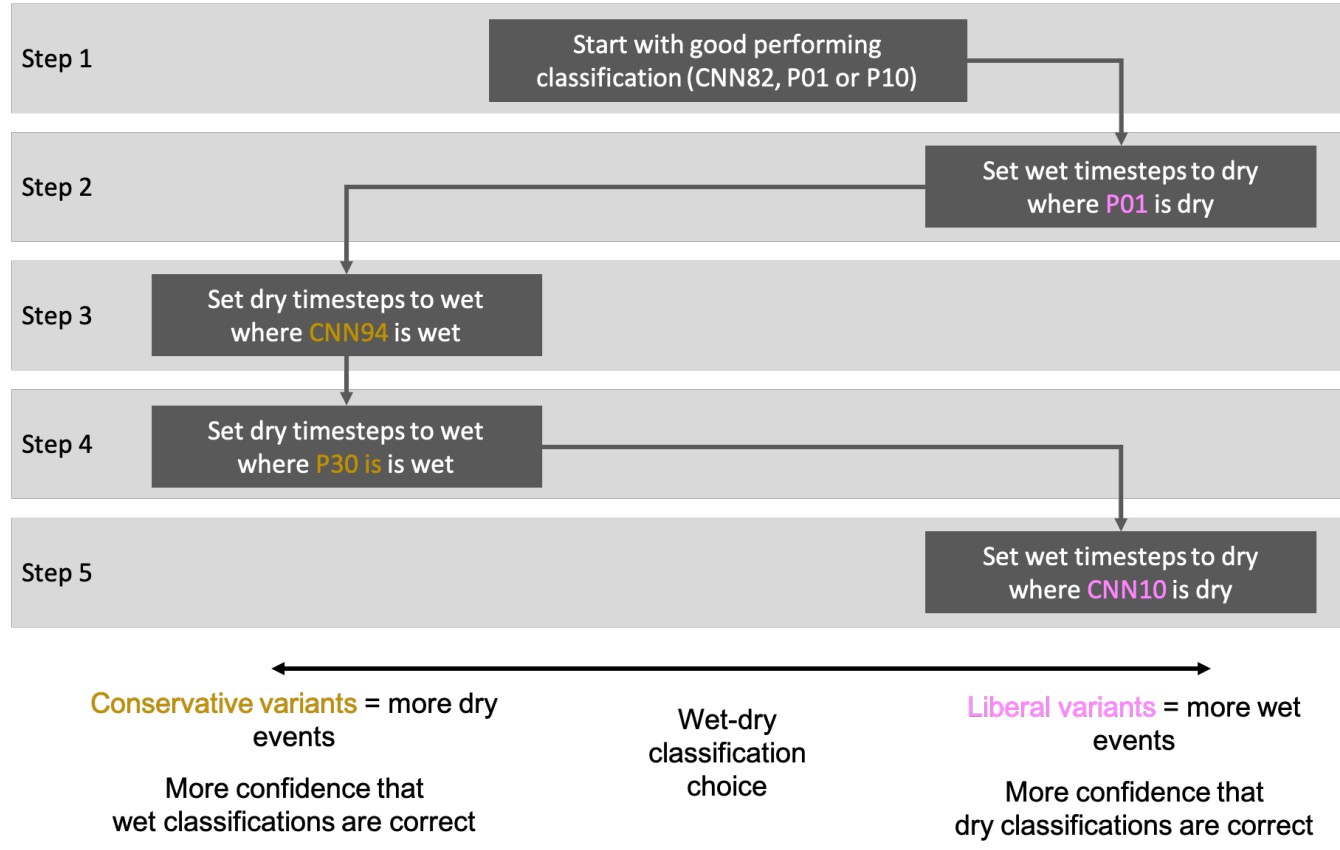

**Figure 3.** Flow chart for building a combination based on TSB and ADB rain event detection methods This step-wise approach starts with an already well-performing classification as a starting point. Then, during each step, time steps that have a high likeliness of being dry (or we) from a very liberal (or conservative) classification from the two methods used are added to the previous classification. If *PC* is used as a starting point, Step 2 and 4 also use *PC*. The same is true for *PC-Ph*. For *CNN82* as a starting point, we computed one variant with *PC* and one with *PC-Ph*. An algorithmic description of this flow chart can be found in Fig. A1.

removed default values and outliers that were outside the range [-10,40] dB for TSL and [-99,0] dB for RSL from the analysis. The total path loss along the CML (TL) was then calculated as TSL minus RSL. We interpolated gaps in TL time series of up to five minutes to obtain a more continuous data availability.

For the rain event detection, we used two TSB methods, two ADB methods with four different thresholds as well as six combinations of TSB and ADB methods. As SEVIRI is only available with a 15-minute resolution, we resampled their classification to a 1-minute resolution with a forward-fill. We computed the following processing steps for each rain event detection method on a 1-minute basis to derive individual rainfall estimates. The baseline attenuation was dynamically identified from the preceding dry period of each rain event to derive the rainfall-induced attenuation along the path. The baseline was determined by the last dry time step of the TRSL time series and was set to be constant during the rain event. As Water on the CML antenna covers can lead to additional attenuation, this so-called wet antenna attenuation (WAA) effect has to be compensated for. We used the WAA correction from Leijnse et al. (2008). In this physical approach, the WAA depends on antenna cover properties (refractive index and thickness), microwave frequency, and rain intensity. We used the parameters given by Leijnse et al. (2008). The rainfall rate was derived from WAA-corrected attenuation using the k-R relationship. The parameter settings for the k-R relation were taken from ITU recommendations (ITU-R, 2005). From a purely practical point of view, we only evaluated data from the first of the two available sublinks.

For comparison with the reference, the binary classifications from the TSB methods and all resulting rain rates were resampled to a 15-minute resolution. We believe that a resampling to 15 minutes and the fact that *RADKLIM-YW* is adjusted to rain gauges suffices to overcome the potential temporal mismatch between radar and ground-based CML observations. The evaluation with a 15-minute resolution might lead to worse results than one on an hourly basis. But for future applications of SEVIRI products for CML processing in regions with sparse reference data like Sub-Saharan Africa, this high temporal resolution is of advantage.

## 3.3 Statistical measures

The radar rainfall estimates at a 15-minute resolution serve as the ground truth for the computation of the scores listed below. For binary classification scores, the ground truth is considered wet if the path-averaged radar rain rate ($r_{ref}$) along the CML path is larger than $0.1 \text{mmh}^{-1}$. The Pearson correlation coefficient (PCC) is used to evaluate the quality, in terms of the linear correlation, of different CML rainfall estimates ($r_{cml}$) derived by using the proposed methods for rain event detection.

$$PCC = \frac{\sum (r_{ref} - \overline{r_{ref}})(r_{cml} - \overline{r_{cml}})}{\sqrt{\sum (r_{ref} - \overline{r_{ref}})^2}\sqrt{\sum (r_{cml} - \overline{r_{cml}})^2}}, \tag{1}$$

where the $\overline{r}$ indicates the mean of a quantity. The relative bias (RB, Eq. 2) is then used to measure an over-, or underestimation that cannot be derived from the PCC.

$$RB = \frac{\sum (r_{cml} - r_{ref})}{\sum r_{ref}}, \tag{2}$$

Binary classification scores are based on the confusion matrix (see Eq. 3):

$$240 \quad \begin{pmatrix} TP & FP \\ FN & TN \end{pmatrix} = \begin{pmatrix} WET/wet & DRY/wet \\ WET/dry & DRY/dry \end{pmatrix} \tag{3}$$

with uppercase and lowercase denoting observed events and predictions, respectively. The correctly assigned wet time steps are called True Positives (TP) and the correctly assigned dry ones are True Negatives (TN). False Positives (FP) represent the number of time steps where rain event detections are incorrectly assigned wet time steps and False Negatives (FN) represent the number of incorrect dry time steps. The confusion matrix fully explains the performance of a classifier, but since the

interpretation of four individual numbers is not straightforward, the computation of additional scores is necessary. The first simplification is to reduce it to the pair of True Positive Rate (TPR, Eq.4), that is the probability of a positive event being predicted positive, and False Positive Rate (FPR, Eq.5) also called false alarm rate. We include both scores since the importance of a high TPR or FPR may be weighted differently depending on the application of the CML rainfall rates.

$$TPR = \frac{TP}{(TP + FN)} \tag{4}$$

$$FPR = \frac{FP}{(FP + TN)} \tag{5}$$

In this manuscript, we focus on improving the overall performance of rain event detection and thus use the Matthews Correlation Coefficient (MCC, Eq.6) which is more robust to influences of the skewed distribution of wet and dry classes (the ratio is roughly 1:20). The MCC is high only if the detection performance for both wet and dry classes is high.

$$255 \quad MCC = \frac{(TP \cdot TN - FP \cdot FN)}{\sqrt{(TP + FP)(TP + FN)(TN + FP)(TN + FN)}} \tag{6}$$

The classifier Accuracy (ACC, Eq.7) is used to analyze the performance for the different rainfall intensity classes and gives the percentage of the time steps in the intensity class that were detected as wet (dry for the dry class).

$$ACC = \frac{(TP + TN)}{(TP + FP + FN + TN)} \tag{7}$$

## 4   Results

In this section, we first compare the ADB products derived from Meteosat-SEVIRI with the weather radar reference to assess their suitability as wet-dry indicators within the processing of CML data. Subsequently, we analyze the relative performance of ADB and the combination of ADB and TSB methods with respect to the performance of established TSB methods. Then, we analyze the performance regarding different rain intensity classes and investigate the influence of different rain event detection methods on the performance of individual CMLs. All presented rain event detection methods were used in combination with

the CML processing routine described in Sec. 3.2 and the resulting data sets were compared to the path-averaged weather radar reference on a 15-minute basis. All scores computed for all methods and the full dataset are additionally shown in (see Tab. B1).

### 4.1 Rain event detection performance of the *PC* and *PC-Ph* SEVIRI product

To assess the quality of the SEVIRI products, they can be compared directly to the weather radar reference. To overcome the issue of their different grid sizes and to asses their quality for using them for CML processing, we compared SEVIRI and *RADKLIM-YW* data as path averages along the CML paths. To compare the PCC of the different SEVIRI-based products and the applied thresholds, processing the CML rainfall rates according to Sec. 3.2 was done using the SEVIRI wet-dry indicator. Fig. 4 compares the performance of the *PC* and *PC-Ph* products according to the TPR, FPR, MCC, and PCC scores. The highest TPR (0.83 during the day and 0.9 during the night) was achieved by the lowest threshold for *PC* and *PC-Ph* (*P01*), which also showed the highest FPR (0.14 day and 0.22 night). For *PC*, the TPR and FPR were increased during the night, but for *PC-Ph*, only the FPR was significantly increased during the night, except for the *P30* version which also showed a higher nighttime TPR. Both, the TPR and FPR decreased with increasing threshold which is due to the decreasing number of positive predictions. The ratio of $TP + FP$ and $TP + FN$ describes how many more timesteps a method predicted as wet, compared to the reference. Accordingly, *PC01* showed 195% (151% for *PC-Ph01*) more wet time steps than the radar data and *PC30* showed the opposite behavior with 36% (52% for *PC-Ph30*) less wet time steps than the reference (see Tab. B1).

The MCC shown in Fig. 4 which measures the overall classification performance showed that the *P10* threshold yields the best results for *PC* during day and night. The best results for *PC-Ph* were achieved by the *P10* threshold during daytime and by the *P01* threshold during nighttime.

Fig. 7b) shows the accuracy of *PC* and *PC-Ph*. The accuracy increases with a higher threshold whereas the relative bias shown in Fig. 7d) becomes more negative with higher absolute values suggesting an increasing underestimation compared to the radar. This is due to an increasing amount of TN and FN predictions.

When computing CML rainfall rates based on the *PC* and *PC-Ph* wet-dry labels, the highest PCC was achieved using the *P01* threshold with values around 0.72. *P10* showed only marginally lower scores. When higher thresholds were applied, the performance decreased more severely. Overall, *PC* showed slightly higher scores than *PC-Ph* and the performance during daytime was equal to or better than the performance at nighttime.

According to this analysis, *P01* and *P10* were the most promising thresholds to apply for SEVIRI-based, pure ADB wet-dry detection. Both, *PC* and *PC-Ph* show good classification and regression scores for these thresholds. Because of the bad PCC, *P30* is not suitable as a threshold for a standalone ADB method and we did not analyze its performance any further. However, we did use it for wet labels with high confidence in the combination algorithm.

### 4.2 Comparison of ADB (*PC* and *PC-Ph*), TSB, and combined rain event detection methods

Due to the increased number of products that are evaluated in this subsection, we limit the analysis to using only the MCC as a measure of classification performance since it gives the best summary of the confusion matrix in one scalar value. Fig.5 shows an analysis that was carried out for the two TSB methods *RS* (grey) and *CNN* (green) and for six ADB methods using the three lowest thresholds (.1, 10 and 20) of *PC* (blue) and *PC-Ph* (orange). The SEVIRI products, *PC* and *PC-Ph*, performed similarly well as the *CNN* and *RS* methods during daytime with MCC scores ranging from 0.49 to 0.51. The best performance

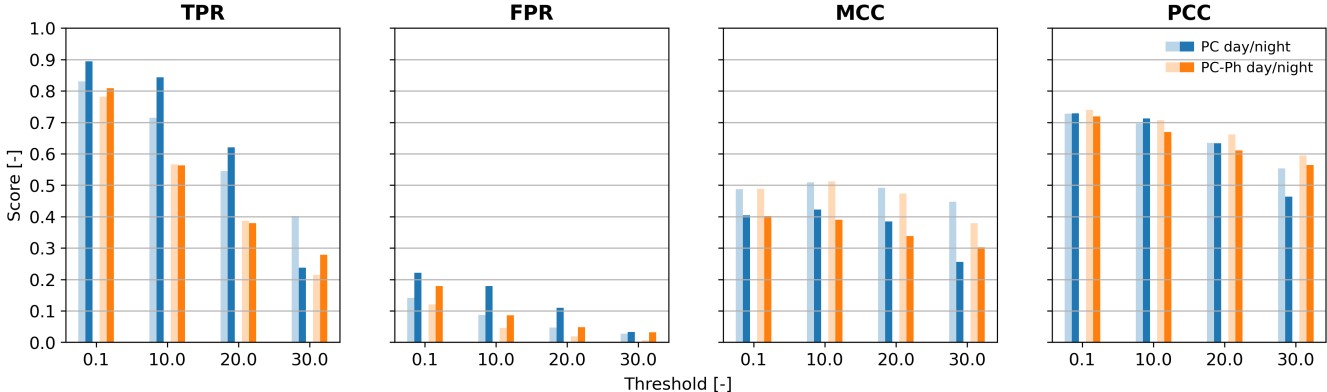

**Figure 4.** Performance metrics of the binary rain event classification (TPR, FPR, and MCC) and the rainfall rates (PCC) of the *PC* (blue) and *PC-Ph* (orange) products compared to the radar reference. The results of each score are presented for different thresholds (x-axes) and split into day (light colors) and night (dark colors).

during daytime was achieved by *PC-Ph10* and the worst by *PC-Ph20*. During the nighttime, the CNN outperformed the ADB methods, while the standard deviation approach showed a similar performance to *PC* and *PC-Ph* except for the higher *P20* threshold which showed the worst performance. Probabilities of 20 % and above (not shown here) showed even worse results. At night, the results of *RS* were worse than during the day. One reason could be dew formation on the CML antenna cover causing an increasing signal attenuation resulting in increased standard deviation which leads to false classification as wet. The *CNN* did not show differences between day and night suggesting that it coped better with this issue. The *SEVIRI*-based products also showed slightly worse results during nighttime, but this was likely due to the lack of three *SEVIRI* channels in the VIS and NIR at night.

SEVIRI-based rain event detection with probabilities up to 10 % provided a similar performance as *RS* and as *CNN* during the daytime, but performed worse than *CNN* during the night.

Fig. 6 shows an example time series of TL, TP, FP, and FN classifications, and rainfall rate of the *PC10* and *CNN* methods as well as the combined method *PC10-combined*. It can be seen that the choice of rain event detection method has an impact on the resulting rainfall rate. The combined method improves the classification performance and correlation of rainfall rates compared to the pure ADB and TSB methods.

In addition to the individual TSB and ADB methods, Fig. 5 shows six different combined methods (purple and red colors) that are composed of either *cnn*, *PC01*, or *PC10* as a starting point, and done for either *PC* or *PC-Ph*. As before *PC* is shown on the left and *PC-Ph* on the right.

The main difference among the combinations is that *PC* data performed better than *PC-Ph* at daytime and similar at nighttime. *CNN* and *PC(-Ph)10* were the best performing TSB and ADB methods and were, therefore, compared to the combined products. *PC10-combined* showed the largest overall improvements with an MCC of 0.59 compared to 0.52 (*CNN*) and 0.52

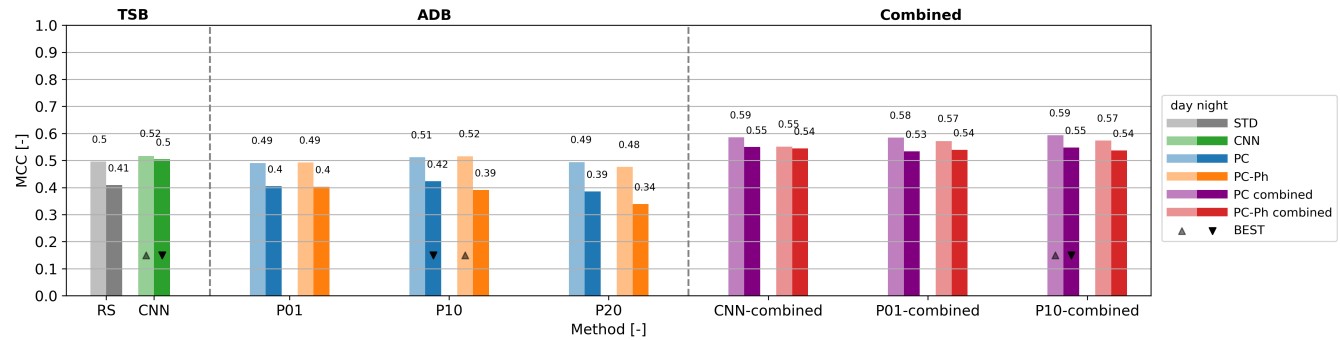

**Figure 5.** Matthews correlation coefficient (MCC) of pure TSB, ADB, and combined methods. *P01*, *P10*, and *P20* refer to the respective *PC* and *PC-Ph* thresholds. The results are split into day- (light colors) and night-time (dark colors). The best performing day and night products of each group are marked with a triangle.

(*PC-Ph10*) at daytime and 0.55 compared to 0.50 (*CNN*) and 0.41 (*PC10*) at nighttime. *PC10-combined* also showed the lowest relative bias of -2.1% compared to *PC01-combined* with 2.6% and *CNN-combined* with -10.8% (see Fig. 4 d). The rel. bias was also lower than that of pure TSB or ADB methods. In the following, we concentrate on *PC10-combined* to evaluate the performance gain using a combination of ADB and TSB methods in more depth, for different intensity classes and individual CMLs.

*PC10* showed the best performance when combined with the *CNN* method (described and shown in the next section) and is thus used as a new ADB method for all final results in the following sections.

## 4.3 Performance of rain event detection methods for different rain intensity classes

Fig. 7 shows the accuracy and relative bias for the TSB, ADB, and combined methods. The relative bias is, for all individual intensity classes, a percentage of the average reference rainfall rate (mean over all classes) such that Fig. 7 c) is the sum of the classes shown in Fig. 7 d).

It can be observed that for all methods the accuracy increased with increasing intensity and that the *CNN* performed better than *RS* for all intensities which confirms the results of Polz et al. (2020) for the same CML network, but a different period. Therefore, we omit *RS* in the remaining results section. Overall, all rain event detection methods lead to an underestimation of light to heavy rainfall which was only partly compensated by an overestimation from false positives (dry class).

*PC* performed better than *PC-Ph* for light to moderate intensity and similar for heavy intensity, but had a lower accuracy for the dry class suggesting more false positives for *PC* when the same threshold was chosen.

Although more methods are shown, we will focus on the best TSB (*CNN*), ADB (*PC10*), and combined (*PC10-combined*) methods from here on to sharpen the analysis.

For the lowest class light1 we observed large differences in accuracy with 60.1% (*CNN*), 73.2% (*PC10*), and 77.8% (*PC10-combined*), while the accuracy was similar for the highest class heavy with 93.7% (*CNN*), 91.0% (*PC10*), and 94.3% (*PC10-*

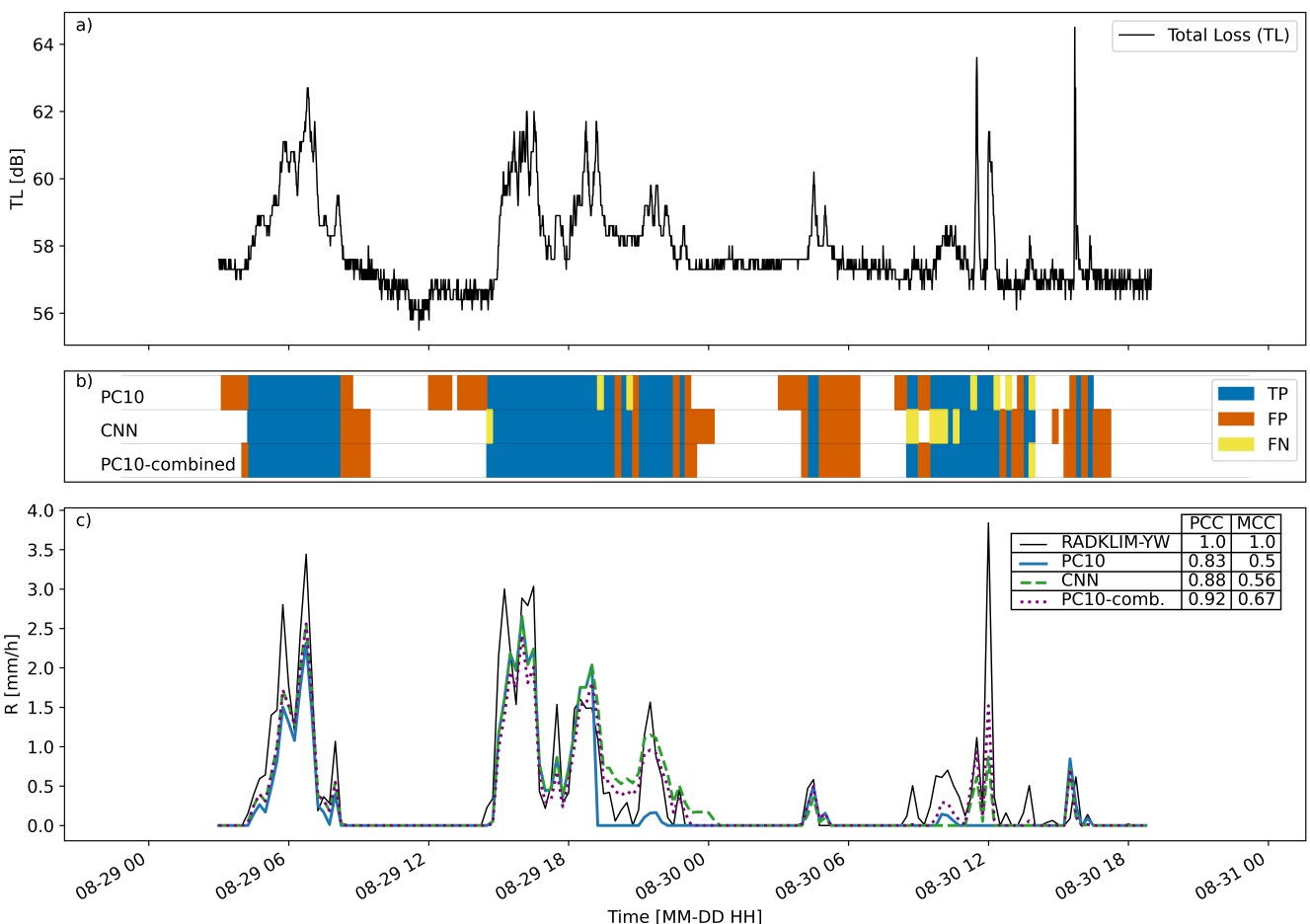

**Figure 6.** Example time series of total loss (TL) (a), classifications compared to the reference (b), and rainfall rates (c). The scores provided in the bottom legend are computed for data from the CML and period shown in this figure.

*combined*). The TSB and combined methods showed a similar dry accuracy which was higher than for the ADB methods.

The positive relative bias due to FPs (Fig. 7c) dry class) is similar for *CNN* (11.3%), *PC10* (8.9%), and *PC10-combined*

(12.7%). The relative bias for the light, moderate, and heavy classes is negative (underestimation) for all methods but has a smaller absolute value for *PC10-combined*. For example, the bias for the light2 class is -7.1% (*CNN*), -8.4% (*PC10*), and -4.4% (*PC10-combined*). Overall, *PC10-combined* (-2.1%) shows a much lower bias than the TSB and ADB methods *CNN* (-11.5%) and *PC10* (-20.8%).

Fig. 8 shows histograms of the occurrence and accumulated rainfall amount of *CNN*, *PC10*, and *PC10-combined* using loga-

rithmic bin widths to be able to visualize differences for all intensity classes. *PC10-combined* showed the highest count of TPs and the lowest count of FNs. Despite this, the number of FPs was lower than for *PC10*. *PC10* showed a similar TP count as *CNN*, but also has the highest amount of FPs. The *CNN* was the most conservative method with the lowest amount of FPs. The

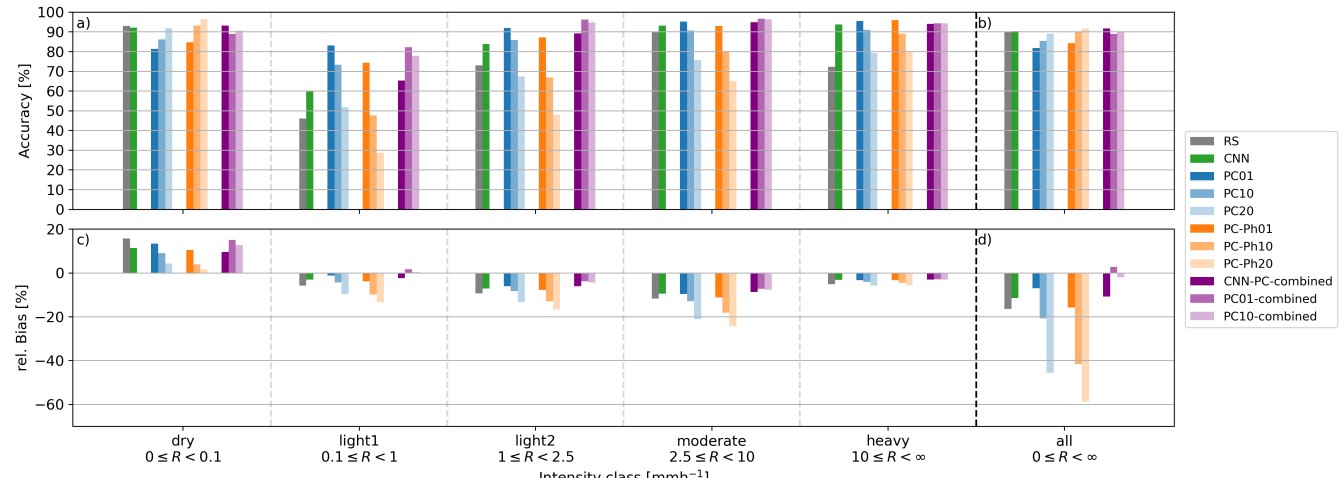

**Figure 7.** Accuracy (top) and relative bias (bottom) of selected TSB, ADB, and combined methods (see legend). Both scores are shown for different intensity classes (x-axis) as defined by the radar reference. The accuracy is the percentage of correctly detected rain events in this intensity class. The relative bias in each class is the sum of all errors in one class as a percentage of the total rainfall of the full reference data. This way, the five leftmost classes add up to the 'all' class on the right.

average rainfall amount per count per CML was different for TPs and FPs. TPs occurred for higher rainfall rates and, therefore this average was 0.43 mm (*CNN*), 0.39 mm (*PC10*), and 0.40 mm (*PC10-combined*). FPs occurred for lower rainfall rates and,

therefore the average was 0.09 mm (*CNN* and *PC10* combined) and 0.07 mm (*PC10*). The average missed rainfall per FN, as measured by the radar reference, was 0.16 mm (*CNN*), 0.21 mm (*PC10*), and 0.15 mm (*PC10-combined*).

To analyze the confidence of individual methods for light, moderate, and heavy rainfall predictions, we computed the probability of a positive prediction to coincide with the reference (i.e. the ratio $\frac{TP}{TP+FP}$). For the light1 class, *PC10-combined* was the most confident method with 0.56 compared to 0.55 and 0.53 (*CNN* and *PC10*). For the light2 and moderate classes, *PC10*

was the most, and *CNN* was the least confident method. For the heavy class, *CNN* was the most, and *PC10-combined* was the least confident method.

## 4.4 Influence of chosen rain event detection method on individual CMLs

The results so far were based on metrics that we computed using all CML data. The performance of individual CMLs might, however, differ from this mean behavior, and systematic differences between CMLs that compare well or badly with the

365 reference are possible. Therefore, scatterplots of the MCC and PCC calculated for each CML individually are shown to compare *CNN* and *PC10* to *PC10-combined* in Fig. 9. For the majority of CMLs, the MCC and PCC could be improved by using *PC10-combined* instead of the best TSB (*CNN*) and ADB (*PC10*) method. The average of the MCCs using *PC10* (resp. *CNN*) was increased from 0.44 (0.51) to 0.56. The CMLs with the worst MCC from *CNN* were improved most when using the combination. For PCC the improvement was smaller (from 0.75 (0.78) to 0.79) but still affected more than 80% of all CMLs.

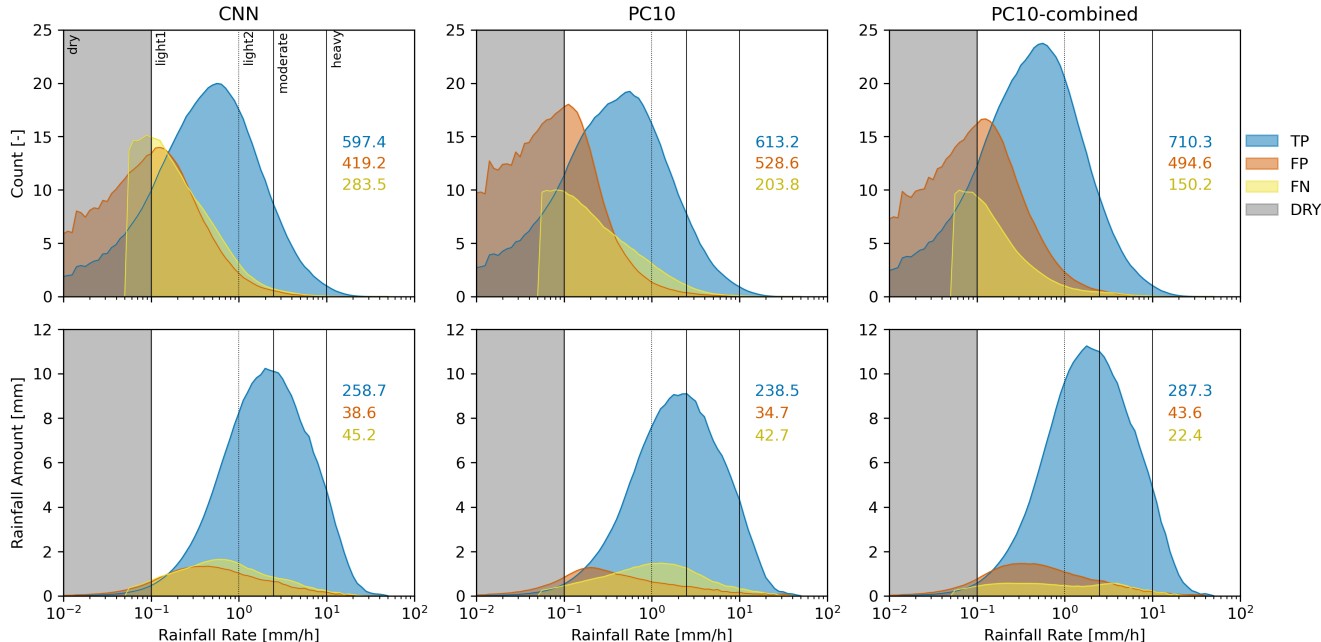

**Figure 8.** In the top panels, the number of FP, TP, and FN time steps with a specific reference rain rate are compared for *CNN* (left), *PC10* (middle), and *PC10-combined* (right). The respective amount of rainfall is shown in the bottom row. All quantities are an average per CML. The rain intensities for FP and TP are estimated by the CML, while the rain intensities for FN, where the CML rainfall rate is zero, are taken from the reference. The numbers in each panel are color-coded according to TP, FP, and FN and show the integrated amount for each curve.

While *PC10-combined* compared to *PC10* improved the PCC by up to 0.36 for individual CMLs, the largest improvement from *CNN* to *PC10-combined* was 0.15. CMLs with the worst PCC could not be improved by using *PC10-combined*. Compared to the overall PCCs of *PC10*, *CNN*, and *PC10-combined* shown in Tab. B1, the mean of all CML's PCCs is slightly higher for all three products. This means that individual CMLs have a higher linear correlation with the reference than the full dataset. Therefore, one can assume that individual CMLs show different biases that could not be compensated by any of the

methods.

## 5   Discussion

### 5.1   Suitability of PC and PC-Ph products as wet-dry indicators for CML data

The performance of *PC* and *PC-Ph* as wet-dry indicators for CML data was analyzed in Sect. 4.1 and compared to TSB methods in Sect. 4.2. The results showed that the classification scores were only slightly lower than for the TSB methods.

In general, the probability threshold had a larger influence on the rain event detection than the differences between *PC* and *PC-Ph*. Both products showed better performance for smaller probability thresholds. The linear correlation (PCC) of derived

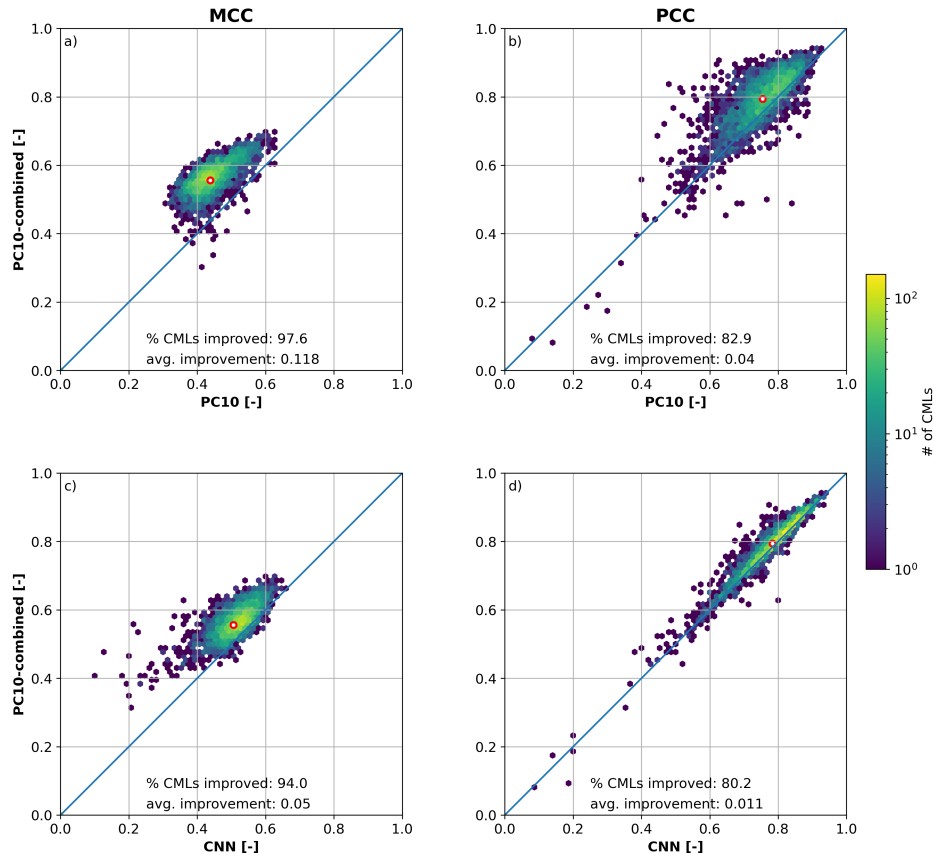

**Figure 9.** Scatter density comparison of the combination of ADB and TSB methods (PC10-combined) with pure ADB (*PC10*, top row) and TSB (*CNN*, bottom row) methods using the Matthews correlation coefficient (MCC, left column) and the Pearson correlation coefficient (PCC, right column). Each score is computed individually for each CML. The red point is the mean of the respective metrics of all CMLs per product.

CML rainfall rates with the reference was highest, when the threshold was lowest, which is due to the lower impact of FPs on PCC compared to FNs. This is surprising because a more liberal classification than using PC01 was not possible. Usually, extremely liberal or conservative methods are assumed to perform badly, which was the case for conservative thresholds *P20*
and higher. For the SEVIRI products, this does not seem to be the case. One reason might be that the cloud mask, which both products use, has good performance over Europe and therefore has a high probability of true dry detection, resulting in a low number of FPs in general and thus also for low thresholds.

In summary, *PC* and *PC-Ph* products perform surprisingly well as wet-dry indicators for CML data. Since these ADB methods are independent of the CML data, they can significantly improve the rain event detection step for noisy CMLs where erratic
fluctuations hamper TSB methods. One limitation of the approach based on SEVIRI data is that the temporal resolution can be more than one magnitude lower than the resolution of the CML data (e.g. 15 min. vs. 1 min.). However, the recently launched

Meteosat Third Generation (MTG) satellites with the Flexible Combined Imager (FCI) will improve the temporal resolution from 15 minutes of MSG SEVIRI to 10 minutes.

## 5.2 Performance of a combination of TSB and ADB rain event detection methods compared to TSB-only and ADB-only methods

The combined method was able to outperform both, pure ADB, and TSB methods in detecting rain events. It was able to make a good trade-off between conservative and liberal methods used in the combination, which was shown by the superior MCC scores. Fig. 6 illustrated how such an improvement can be achieved, for example by reducing FNs that lead to a reset of the baseline at a higher level that ultimately leads to too small rainfall rates for *PC10*.

The choice of a starting method for the combination only had a small impact on the results, because most, but not all, initial predictions are overwritten in the combination process. The best MCC and lowest bias were achieved by *PC10-combined* which is why it was chosen for a more detailed comparison of scores for individual CMLs.

*PC10-combined* improved the MCC and PCC scores for the vast majority of all CMLs and only single outliers achieved better scores using the pure ADB or TSB method. Thus, we conclude that the combination algorithm is robust against varying CML behavior. A limitation of the proposed combination is that **CNN** relies on one-minute instantaneously sampled CML data. However, a similar combination of ADB with other TSB methods that can handle for example the common 15-minute "min-max" sampling should be easily applicable following the logic presented in 3.

## 5.3 Performance differences between day and night

We observed a notable classification performance drop during nighttime for *PC* and PC-Ph, but also for *RS*. For *PC* and *PC-Ph* this day and night difference is very likely due to the missing SEVIRI channels in the VIS range at night. The calculation of microphysical variables is mainly based on these channels. Their absence consequently limits the reliability of all derived precipitation products. According to Fig. 4), this effect significantly increases the rate of positive predictions.

While *CNN* was able to perform equally well during nighttime, the other TSB method *RS* showed a decreased performance. One possible explanation could be the formation of dew on the antennas during nighttime that can regularly be observed in CML time series as a slowly increasing (after sunset) and decreasing (after sunrise) attenuation. The more sophisticated pattern recognition algorithm of the *CNN* method seems to be able to correctly classify these periods as dry. The combined methods utilized the high confidence of the *CNN* and reduced the day and night difference.

However, the PCC results for *PC* and *PC-Ph* showed that a decrease in MCC during the night did not lead to a worse correlation of derived rainfall rates.

## 5.4 Performance for different rainfall intensity classes

Our results confirmed that the detection performance is much higher for moderate and heavy compared to light rainfall. This was already shown for TSB methods by Polz et al. (2020) and is now also confirmed for the ADB methods based on *PC* and

*PC-Ph*. The TSB methods have to distinguish between rainfall signal and noise which can become similar for low rainfall intensities. The ADB method using SEVIRI data suffers from its indirect measurement principle and may have difficulties in distinguishing between precipitation and non-precipitating clouds, particularly for light rain.

The contribution of the different intensity classes to the overall relative bias in Fig. **??** shows that the overestimation due to FPs in the dry class is smaller than the underestimation in the positive intensity classes where FNs are one major factor for missed rainfall. This way, all methods except *PC01-combined* show an overall negative bias. One additional explanation for an overall negative bias could be a too-large compensation for WAA. Tiede et al. (2023) described strong WAA fluctuations during long-lasting rain events, a fact that is not considered in the WAA compensation method we chose (or any other available method). Therefore, there is some uncertainty in WAA compensation which influences the bias.

Although *PC10* was more liberal, i.e. generally favoring positive predictions, the *CNN* was more confident in the prediction of heavy rainfall. The combined method increased the performance on low rain rate TPs compared to the TSB method, because of the higher confidence of the ADB method with only a small increase in the count and rainfall amount through FPs. Therefore, we confidently claim that *PC10-combined* was able to improve the detection performance for the dry and all positive intensity classes.

## 6    Conclusions

In this study, we aimed to address the questions of whether satellite-derived precipitation products are suitable indicators for rain event detection in CML data with respect to day/night differences, different rain intensities, and whether combinations of TSB and ADB methods show an additional added value. We achieved this by using *PC* and *PC-Ph* from MSG SEVIRI in an ADB rain event detection and compared the results to two TSB rain event detection methods. Then, we combined the most promising variants in such a way that the most confident detection method was used for any given time step.

The results clearly show that *PC* and *PC-Ph* products from MSG SEVIRI can be used for the detection of rain events in CML attenuation time series. They performed almost as well as the TSB methods during daytime and worse than *CNN* during nighttime. Minor differences between the SEVIRI products *PC* and *PC-Ph* exist, but the chosen threshold of precipitation probability dominated the overall behavior. An improved *PC-Ph* product is available since April 2022 potentially making its application in rain event detection even more attractive (personal communication with NWC SAF).

However, the performance of ADB methods based on SEVIRI was lower at night than during the day due to the lack of the three SEVIRI channels (VIS and NIR). Since *CNN* did not show a decrease in quality at night, it would be logical to vary the rain event detection for day and night time. We did not aim for such a temporal variation to avoid inconsistencies in the resulting time series. Compared to the ADB and *CNN*, the *RS* method does not show particular advantages, except for the easy application.

The quality of the rain event detection methods clearly depended on the rain intensity, with a better performance for moderate and heavy rain than for light rain. For flood forecasting light rain is often negligible but this is not the case for water balance analyses. Additionally, the increasing threat of droughts in the context of climate change also requires a high-quality represen-

tation of light rain. Low rainfall intensities show a large potential for improvement and major differences in this study were also obtained there.

The effort to use ADB is larger than for TSB methods because the processing of additionally needed satellite data can be time-consuming. However, the global availability of the data allows for the unified processing of CML datasets from differ-
ent countries and we assume that extensive re-calibration is not needed. Stepwise combinations with TSB methods that need to be adjusted to the characteristics of the CML data, as presented here, do need re-calibration and increase the effort. The shown additional improvements by combinations are promising and justify the effort. There is a multitude of possibilities when combining different rain event detection methods. Using methods like the nearby link approach or using different thresholds depending on data quality or rainfall intensity is also possible in future applications.

In principle, combinations of multiple TSB methods are possible and may lead to improvements. However, we recommend applying a combination of TSB and ADB methods to exploit the advantages of both approaches: TSB methods are easy to apply and provide precise results where the separation of noise and rainfall signals is obvious. Whereas, ADB methods show a better performance for noisy or unstable CML time series due to their independence from the actual CML signal. Burkina Faso for example has only a few rainfall stations, but several hundred CMLs. Most CMLs outside the capital Ouagadougou
are long (>20km) and use frequencies around 7 GHz. Their time series are quite noisy and show large fluctuations from other sources than rain. With abundant information from geostationary satellites and the methods presented in this work, we expect to extract useful ground-based rainfall information from such CMLs in this region with scarce rainfall data.

*Data availability.*   CML data were provided by Ericsson Germany and are not publicly available. RADKLIM-YW was provided by the German Weather Service (DWD) and are publicly available.: https://opendata.dwd.de/climate_environment/CDC/grids_germany/5_minutes/radolan
(last access: 28 July 2023; DWD CDC, 2023)

The PC and PC-Ph products were provided by Llorenç Lliso and José Alberto Lahuerta who are affiliated with NWC SAF. Recent data is shown at https://www.nwcsaf.org/web/guest/nwc/geo-geostationary-near-real-time-v2021 (last access: 20 December 2023) and long-term records must be requested individually from NWC SAF.

## Appendix A:  Additional figures

## Appendix B:  Additional tables

*Author contributions.*   AW, CC, MG, and JP designed the study layout, and AW carried it out with the contribution of CC, MG, and JP. Data and related support was provided by LL and JAL. AW prepared the article with contributions from all co-authors. MG and JP prepared the revised article with contributions from CC.

---

**Algorithm 1**

Combination of TSB and ADB methods. Example of CNN-PC-combined.

---

1: t ← CNN82
2: **if** PC01 is dry **then**
3:     t ← dry                          ▷ high confidence in dry predictions
4: **if** CNN94 is wet **then**
5:     t ← wet                          ▷ high confidence in wet predictions
6: **if** PC30 is wet **then**
7:     t ← wet                          ▷ high confidence in wet predictions
8: **if** CNN10 is dry **then**
9:     t ← dry                          ▷ high confidence in dry predictions

---

**Figure A1.** Algorithm describing the combination *CNN-PC-combined* of *PC* and *CNN* starting with *CNN82*. A given time-step of CML data that needs to be processed is denoted by $t$. The different wet-dry methods give a wet or dry prediction for this time-step as described in Sect. 3.1.1.

**Table B1.** Confusion matrix values TP, FP, TN, and FN as well as classification scores ACC and MCC and regression scores PCC and RB for all considered methods.

| Group | Method | TP | FP | TN | FN | ACC | MCC | PCC | RB |
|---|---|---|---|---|---|---|---|---|---|
| Time series based (TSB) | | | | | | | | | |
| | RS | 2086257 | 2761086 | 36939753 | 1487374 | 0.902 | 0.449 | 0.657 | -0.165 |
| | CNN | 2511242 | 3125653 | 36575186 | 1062389 | 0.903 | 0.51 | 0.742 | -0.115 |
| Aux. data based (ADB) | | | | | | | | | |
| | PC01 | 3103560 | 7412469 | 32288370 | 470071 | 0.818 | 0.438 | 0.73 | -0.07 |
| | PC10 | 2809793 | 5525566 | 34175273 | 763838 | 0.855 | 0.452 | 0.707 | -0.208 |
| | PC20 | 2100523 | 3258161 | 36442678 | 1473108 | 0.891 | 0.423 | 0.636 | -0.455 |
| | PC30 | 1136635 | 1205697 | 38495142 | 2436996 | 0.916 | 0.35 | 0.509 | -0.719 |
| | PC-Ph01 | 2860243 | 6112184 | 33588655 | **713388** | 0.842 | 0.439 | 0.731 | -0.159 |
| | PC-Ph10 | 2029703 | 2703467 | 36997372 | 1543928 | 0.902 | 0.441 | 0.689 | -0.417 |
| | PC-Ph20 | 1376271 | 1408014 | 38292825 | 2197360 | 0.917 | 0.392 | 0.637 | -0.587 |
| | PC-Ph30 | 893173 | **815028** | **38885811** | 2680458 | **0.919** | 0.324 | 0.58 | -0.711 |
| Combined | | | | | | | | | |
| | CNN-PC-combined | 2676762 | 2727541 | 36973298 | 896869 | 0.916 | **0.566** | 0.746 | -0.108 |
| | PC01-combined | **3122738** | 4379019 | 35321820 | 450893 | 0.888 | 0.555 | 0.742 | 0.026 |
| | PC10-combined | 3010509 | 3762008 | 35938831 | 563122 | 0.9 | **0.566** | 0.743 | -0.021 |
| | CNN-PC-Ph-combined | 2529058 | 2588598 | 37112241 | 1044573 | 0.916 | 0.548 | **0.747** | -0.134 |
| | PC-Ph01-combined | 3003963 | 3988729 | 35712110 | 569668 | 0.895 | 0.553 | 0.743 | **-0.017** |
| | PC-Ph10-combined | 2656801 | 2865316 | 36835523 | 916830 | 0.913 | 0.554 | 0.746 | -0.114 |

*Competing interests.* The authors declare that they have no conflict of interest.

*Acknowledgements.* We thank Ericsson for their support and cooperation in the acquisition of the CML data. This research has been supported by the Federal Ministry of Education and Research (Grants 01LZ1904A-C and 13N16432), the Helmholtz Association (Grant ZT-0025) and the German Research Foundation (Grant CH-1785/1-2).

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
