# Peer review of "Improved rain event detection in Commercial Microwave Link time series via combination with MSG SEVIRI data"

_Atmospheric Measurement Techniques, 2023_

## Referee Comment (RC1)

REVIEW OF THE PAPER "Improved rain event detection in Commercial Microwave Link time series via combination with MSG SEVIRI data", AMT 2023-175

**General comment**

The authors compare different algorithms for detecting dry/wet time slots from opportunistic measurements collected by Commercial Microwave Links and, at the same time, they assess whether the wet/dry classifiers can be improved using satellite data collected by MSG SEVIRI. Dry/Wet slot classification is an important step in the processing chain of CML data to extract quantitative precipitation estimates. This is a very specific aspect of the topic of opportunistic sensing of precipitation by CMLs, which I believe fits the scope of AMT. Moreover, I think it has some novel aspects as it is one of the first papers demonstrating the effectivity of data fusion between opportunistic sensors and products from Earth observation satellites.

First the authors check whether SEVIRI products are reliable comparing them with RG-adjusted radar data, chosen as a reference (sec. 4.1). Then, they analyze the performance of several wet/dry classifiers based on CML data only, on SEVIRI data only and on a combination of both. Finally, they assess the impact of different wet/dry classifiers on the performance of CML as quantitative rainfall sensors. The logic sounds correct. However, I think the datasets used, the methods and the results are not well explained. There are several aspects to be clarified: hence, in my view, this contribution needs a major revision before being rated as acceptable for publication.

**Specific comments**

I am kindly asking the authors to address the following specific comments:

- Sec. 3.2 (about methods): I think the wet/dry classification rationale by each of the 3 different types of sensor is not well explained and some information is missing in my view (or maybe I missed it).
    - I suggest to add an itemized list in Sec. 3.2 and explain wet/dry rationales for radar, CML and SEVIRI.
        - Radar data: I missed the way how raw radar data (I guess rainfall intensity) are processed and combined together to build the reference wet/dry time series. Are radar data thresholded above 1 mm/h and then mapped over each CML path? How do you flag a CML as wet/dry from the weighted average of overlapped radar pixels? Is the 1 mm/h threshold applied on the weighted average instead? How did you put together radar and CML time scales (5 min and 1min)? Please clarify.
        - SEVIRI data: there are as many wet/dry time series as are the pNN labels (SEVIRI outputs). For instance, the time series p10 at a certain pixel is generated assuming the pixel wet when it is flagged as p10. Is it correct?
        - CMLs: two wet/dry time series are produced for each link, corresponding to two different methods (RS and CNN). Is that correct?
    - I suggest to add a table with time step and spatial resolution of each sensor. I didn't get which is the radar pixel size. SEVIRI's pixel size depends on the elevation and azimuth of the observed point on the Earth (i.e. Germany). It is not stated which is the range of SEVIRI pixel sizes for Germany. Finally, we can say that the spatial resolution of CML equals their length if no path reduction factor is used.
    - At least in the text, please provide absolute numbers of the populations involved: we are talking about 10,000, 1,000,000 or even longer time series? And what is the total dataset size (number of sensors x number of samples)? Finally, would be good to have some min-max range for positives , i.e. wet link occurrences in the observation period.

- Introduction, lines 61-67: even though they have been proposed for satellite links, it would be good to take a look at the following methods (as dry/wet identification in terrestrial and satellite links are very similar):
    - L. Barthes et al., "Rainfall measurement from the opportunistic use of an earth–space link in the ku band," Atmosph. Meas. Techniq., vol. 6, no. 8, pp. 2181–2193, 2013.
    - C. H. Arslan, et al., "Satellite-link attenuation measurement technique for estimating rainfall accumulation," IEEE Trans. Geosci. Remote Sens., vol. 56, no. 2, pp. 681–693, 2017
    - F. Giannetti et al., "The nefocast system for detection and estimation of rainfall fields by the opportunistic use of broadcast satellite signals," IEEE Aerosp. Electron. Syst. Mag., vol. 34, no. 6, pp. 16–27, 2019
    - B. He et al., "Use of the c-band microwave link to distinguish between rainy and dry periods," Adv. Meteor., vol. 2019, 2019
    - M. Xian et al., "Rainfall monitoring based on machine learning by earthspace link in the ku band," IEEE J. Sel. Topics Applied Earth Observ. Rem. Sens., vol. 13, pp. 3656–3668, 2020.
    - R. Giro et al., "Real-time rainfall estimation using satellite signals: Development and assessment of a new procedure," IEEE Trans. Instrum. Meas., vol. 71, pp. 1–10, 2022.
    - C. Gianoglio, et al., "Rain discrimination with machine learning classifiers for opportunistic rain detection system using satellite micro-wave links," Sensors, vol. 23, no. 3, p. 1202, 2023.
- Sec. 3.3: Performance indicators
    - is it really beneficial to introduce all those indicators? In principle, sensitivity (i.e. TPR according to the authors) and specificity (not considered by the authors) should be almost all we need. MCC can be useful as it is a global indicator combining FP and FN rejection, but it is not as straightforward as the previous two. It's not easy to state how good is an MCC value equal to 0.60-0.62 and how much an increase of MCC by 0.09 and 0.13 (I took these numbers from the abstract) is indeed valuable. Indeed, it is not obvious to make ratings of methods based on the MCC values in Figs. 5 and 7. Finally, please note that the importance or having high sensitivity rather than high specificity methods or vice-versa, depends on how wet/dry classification is used within the CML processing chain. Is it used to calculate the baseline? In this case, not misclassifying wet slots as dry is critical, i.e. sensitivity is the key indicator. I think the way you look at wet/dry classification as a part of data processing will drive the choice of the most significant performance indicator. Some discussion and a better justification of the indicators used is needed, instead of listing formulas and writing definitions.
    - ACC is shown in Fig. 3 (purple bars, I see small differences among bars) and in Fig. 9, first column. PCC is shown in Fig. 3 and Fig. 9 as well. Unless these two highlight different aspects of the confusion matrix than MCC, I think it's simpler and better to show only MCC throughout.
    - The PPV indicator sounds a bit ambiguous to me in the framework of CML data, because it gets low (i.e. poor performance) if either there are a lot of FPs with respect to TPs, but it is also low for a given number of FPs if there are few TPs on the whole. FPs are often produced by factors other than rain, hence they have not in a strict relation with the number of TPs.
    - An interesting investigation would be to assess where errors (FP and FN occurrences) actually are. Are they at the start of an event rather than at the end? Or there are sequences of FPs far from events? Have the authors done such kind of analysis?
- Figures 3,4,5,7: it's not a good idea to put "night" or "day" as labels on the y-axes of those figures. You should put the quantity displayed in the graph instead. Night and day should be placed into a text box in a free space over the figure or as descriptors after subfigure identifiers (e.g. a) , b), etc.) above each graph. Same for the x-axis: for instance, in Fig.3, the authors used different labels for the

x-axis of subfigures a) and b): actually, they are the same axes. And they put TPR (y-axis) on the x-axis. The figures are indeed complex and they aggregate too much information spread in too many dimensions (night/day, percentage of rain occurrence as from SEVIRI products, rain intensity class, statistical indicator). Moreover, those small pictures on the top right of Fig. 3 are not well described: I guess here the TPR is weighted with the accumulated precipitation, that's why it is different from the blue histogram of Fig. 3b. Not straightforward really. It definitely looks too much and too difficult to track. Please simplify, dropping less meaningful dimensions. For instance, in Fig. 3a, TPR shows minor differences between night and day, as the authors state on page 10. Hence, one of the graphs of Fig 3a) can be dropped in my view. Also, I can't see any significant difference or trend between the two different SEVIRI products in Fig 3a.

- Figure: 3: the authors show TPR of SEVIRI-based wet/dry classifier assuming radar-based classifier is the truth.
  - Is this comparison carried out mapping radar and SEVIRI pixels onto CML paths or it is just that the SEVIRI grid has been mapped onto the radar grid?
  - I didn't get the trend of TPR as a function of pNN, where NN is the probability of precipitation in percent (as the authors stated on page. 5), being pNN a SEVIRI product. So I expect TPR to increase with pNN, as SEVIRI wet/dry time series with high pNN have less wet slots than the ones with small pNN, hence a lower probability to incur in FNs. Why is it the opposite? On the other hand, the trend of PPV with pNN in Fig. 3b looks to agree with the feeling that a SEVIRI sample with high pNN is really a wet sample.

- Fig. 4: I have several comments here.
  - on Page 11, Line: 257 the author state: "The distribution of rain intensities and total precipitation amount of eight rain event detection methods is shown in Fig.4". It's way too generic. The explanation in the figure caption does not really help either (it should be improved as well). Finally, the word "count" in the x-axis label of the figure has nothing to do with rain intensities (following the logic of Fig. 3, count is the quantity displayed on the y-axis). So, what did you actually plot in Fig. 4? Is it a comparison between rain intensity/depth estimates across CML paths done by CML (2 methods)/SEVIRI (6 products) vs radar measurements? Or is it the percent difference between CML/SEVIRI wet counts and radar wet counts averaged over the population of a certain intensity class further divided by day and night? By the way, when you described SEVIRI data in Sec. 2.3, you didn't state clearly if precipitation intensity/depth are among SEVIRI products or retrievable from SEVIRI products., so I guess you are talking about counts. Please clarify and correspondingly edit your manuscript and it is really hard to get out of this based on what is written in explanations. Sec. 4.2.1 and in the figure caption.
  - In their explanation of the figure on page 11, the author state that "The SEVIRI-based ADB data sets behaved very similarly to the two TSB data sets RS and CNN". When I see it, TSB data work better than most of SEVIRI products except for the high-intensity class.
  - "a tendency of radar data to underestimate heavy rain intensities" (page 11 to justify CML/SEVIRI overestimate at large intensity) is a dangerous statement in my view. So far, the authors considered radar data as the truth and now they state that they maybe not good in estimating heavy rainfall. The author should provide a convincing evidence based on literature focused on German radar data, rather than citing a paper (Schleiss 2020) that worked over data from other radar networks.

- Results: in Fig. 9., the authors assessed the effectivity of a classifier based on mixing together TSB and ADB methods by performance indicators derived from the confusion matrix. The authors stated in the abstract "The separation of the attenuation time series in rainy and dry periods (rain event detection) is the most important step in this processing and largely determines the quality of the resulting rainfall estimates." So, a reliable classifier will end up in improving CML-based rainfall

estimates. This is not demonstrated, however. It would be really good to see how Pc10all decreases the error on rainfall intensity estimates with respect to Pc10 and CNN through a scatterplot of rainfall intensity/depth as the ones in Fig. 9.

**Technical corrections**

- Page 1, pp. 38-39: "Nevertheless, gauge-adjusted radar data is considered to be the best possible data basis", quite a strong statement if you ask me. I would be happy to state that RG calibrated radar data are considered reliable for estimating precipitation over large areas.
- Page 3, line 65: a full stop is missing
- Page 3, line 72: NWC SAF acronym not explained and written as one single word on following line 84.
- Page 3, line 74: "has carried out analyzes" is misspelled
- Page 5, line 125: "in Thoss" rather than "at Thoss"
- Page 5 line 126: a full stop is missing
- Page 7, line 189: "In step 1, we choose a method (either CML-TSB or SEVIRI-ADB) that shows a good average performance as a first guess"
- Page 7, line 193: Section rather than Chapter
- Page 10, Line 230: "To assess the quality of SEVIRI data it was 230 directly compared to radar data", this statement sounds a bit awkward: a comma is missing at least, or state it better.
- Page 10, Line 237: "over different rain intensity", would state "over different rain intensity values"
- Page 10, Lines 232-33: "TPR shows the percentage of wet RADKLIM-YW time steps per intensity class and precipitation amount, represented by SEVIRI data for different thresholds". Badly written. I am afraid this has to do with the comment I did above about the complexity of figures. I would state it in a simpler way: "Fig. 3a shows the TPR (in percent) of SEVIRI wet/dry classification at day (top) and night (bottom) divided per intensity class and per probability precipitation".
- Page 10, Line 240, "P30 showed the opposite". Cannot get it. p30 just shows that the p30 population of SEVIRI wet slot is closer to the population of RADAR wet slots then the p01 population (as PPV_p30> PPV_p01).
- Page 13, lines 290-292: I guess dew effect is temperature dependent. I am not asking the authors to do such an analysis, but maybe processing data based on local temperature classes would help to clarify this point.
- Page 17, caption of Fig.8: "The rain intensities for FP and TP are estimated by the CML, while the rain intensities for FN are taken from the reference". I cannot get it. Why didn't you take rainfall intensities from the reference (i.e. radar) all the say? Please clarify.

---

## Author Comment (AC1)

**General statement about necessary corrections to the methodology**
We thank the reviewers for their constructive criticism of our manuscript. In the process of revising our manuscript a few changes to the methodology, as presented in the initial submission, have turned out to be inevitable. These changes do impact the presented results, but the most important conclusions remain the same. Here we want to summarize the changes to our methods, how the main results between the old and new versions differ, and what we can conclude from that. In the revised manuscript we will only show the new results from the updated methodology.

The changes to the methodology are as follows:

- In the initially submitted manuscript, detected rain event that, later in the processing did yield a CML rainfall rate smaller than 0.1mm/h were considered "dry". Such periods occur, for example, when the estimated baseline is set higher than or equal to the total loss (TL), or when the WAA estimate reduces the rainfall rate below the selected threshold of 0.1 mm/h. We admit that the previous description of the methodology did not explain this very well. To be in line with the vast majority of studies on rain event detection and to reduce the complexity of the results we now use the "wet" and "dry" labels directly from the output of the rain event detection methods and do not change any of them based on the resulting CML rainfall rate.
- Polarization provided to the wet antenna attenuation estimation method after Leijnse et al. (2008) have been corrected. In the previous version, a horizontal polarization was used in the estimation of WAA for all CMLs. The change affects 3509 CMLs (all that have vertical polarization) and slightly reduces the estimated rainfall amount. The actual retrieval of rain rates via the k-R relation was done with correct polarization data.
- The total amount of CMLs considered in the study was reduced by excluding those that either show a data availability of less than 30% of all time steps or have a constant signal. The amount of CMLs was previously 3901 and is now 3748.

The effect of these changes are presented in Figs. AC1 and AC2 below.

[Figure]

Fig. AC1: Comparison of scores (as shown in old Fig 9) from the old and new updated method. Shown are the pc10 (top row), cnn (middle row), and pc10 combined (bottom row) products as well as the Matthews correlation coefficient (MCC, left column) and the Pearson correlation coefficient (PCC, right column). New scores are on the x-axis and old scores are on the y-axis. The red circle shows the mean along x and y and the average decrease is the mean difference of x and y.

[Figure]

Fig. AC2: Comparison of a revised Figure 9 computed using the old (left) and new (right) methodology. The revised Figure 9 caption is: Scatter density comparison of the combination of ADB and TSB methods (pc10_combined) with pure ADB (pc10, top row) and TSB (cnn, bottom row) methods using the Matthews correlation coefficient (MCC, left column) and the Pearson correlation coefficient (PCC, right column). Each score is computed individually for each CML.

The comparison in Fig. AC1 implies that, while all scores slightly worsen using the new methodology, it affects all compared methods similarly. The largest effect is visible for pc10 which is a pure Meteosat-SEVIRI based method. While the results do change with the new updated methodology, the main conclusions will stay very similar:

1. The PC and PC-Ph products still prove to be suitable as a wet-dry indicator for CML data, although their performance is now (after we update the methodology) worse than the performance of the cnn method.
2. The results vary with rain intensity showing better performance for moderate to heavy rainfall.
3. There are notable differences between day and night with a reduced performance for the SEVIRI products and the rolling standard deviation method during nighttime.
4. A combination of TSB and ADB rain event detection methods outperforms TSB-only and ADB-only methods.

**General comment**

The authors compare different algorithms for detecting dry/wet time slots from opportunistic measurements collected by Commercial Microwave Links and, at the same time, they assess whether the wet/dry classifiers can be improved using satellite data collected by MSG SEVIRI. Dry/Wet slot classification is an important step in the processing chain of CML data to extract quantitative precipitation estimates. This is a very specific aspect of the topic of opportunistic sensing of precipitation by CMLs, which I believe fits the scope of AMT. Moreover, I think it has some novel aspects as it is one of the first papers demonstrating the effectivity of data fusion between opportunistic sensors and products from Earth observation satellites.

First the authors check whether SEVIRI products are reliable comparing them with RG-adjusted radar data, chosen as a reference (sec. 4.1). Then, they analyze the performance of several wet/dry classifiers based on CML data only, on SEVIRI data only and on a combination of both. Finally, they assess the impact of different wet/dry classifiers on the performance of CML as quantitative rainfall sensors. The logic sounds correct. However, I think the datasets used, the methods and the results are not well explained. There are several aspects to be clarified: hence, in my view, this contribution needs a major revision before being rated as acceptable for publication.

**Specific comments**

I am kindly asking the authors to address the following specific comments:

- Sec. 3.2 (about methods): I think the wet/dry classification rationale by each of the 3 different types of sensor is not well explained and some information is missing in my view (or maybe I missed it).
  - I suggest to add an itemized list in Sec. 3.2 and explain wet/dry rationales for radar, CML and SEVIRI.
    - Radar data: I missed the way how raw radar data (I guess rainfall intensity) are processed and combined together to build the reference wet/dry time series. Are radar data thresholded above 1 mm/h and then mapped over each CML path? How do you flag a CML as wet/dry from the weighted average of overlapped radar pixels? Is the 1 mm/h threshold applied on the weighted average instead? How did you put together radar and CML time scales (5 min and 1min)? Please clarify.
    - SEVIRI data: there are as many wet/dry time series as are the pNN labels (SEVIRI outputs). For instance, the time series p10 at a certain pixel is generated assuming the pixel wet when it is flagged as p10. Is it correct?
    - CMLs: two wet/dry time series are produced for each link, corresponding to two different methods (RS and CNN). Is that correct?
  - I suggest to add a table with time step and spatial resolution of each sensor. I didn't get which is the radar pixel size. SEVIRI's pixel size depends on the elevation and azimuth of the observed point on the Earth (i.e. Germany). It is not stated which is the range of SEVIRI pixel sizes for Germany. Finally, we

can say that the spatial resolution of CML equals their length if no path reduction factor is used.

- At least in the text, please provide absolute numbers of the populations involved: we are talking about 10,000, 1,000,000 or even longer time series? And what is the total dataset size (number of sensors x number of samples)? Finally, would be good to have some min-max range for positives , i.e. wet link occurrences in the observation period.

We agree with the reviewer that a more structured way of explaining the rain event detection methods and the individual sensors and datasets (including pixel size, temporal resolution, number of sensors, number of data points, etc.) would be beneficial. Therefore we will add this information as itemized lists or as a table to the revised data and methods section. We also see the need to improve the general explanation of our methodology and evaluation, especially how we use the weather radar data as a reference. We used radar data for the evaluation of binary and rainfall-based information from TSB, ADB, and combined methods in the submitted manuscript. We compared the path-averaged information from CMLs with the radar grid (or the satellite grid), by calculating the average of all radar (satellite) pixels intersected by a CML, weighted with the length of the CML segment in each pixel (explained in L168). We suggest stating this more clearly in the methods section.

- Introduction, lines 61-67: even though they have been proposed for satellite links, it would be good to take a look at the following methods (as dry/wet identification in terrestrial and satellite links are very similar):
  - L. Barthes et al., "Rainfall measurement from the opportunistic use of an earth–space link in the ku band," Atmosph. Meas. Techniq., vol. 6, no. 8, pp. 2181–2193, 2013.
  - C. H. Arslan, et al., "Satellite-link attenuation measurement technique for estimating rainfall accumulation," IEEE Trans. Geosci. Remote Sens., vol. 56, no. 2, pp. 681–693, 2017
  - F. Giannetti et al., "The nefocast system for detection and estimation of rainfall fields by the opportunistic use of broadcast satellite signals," IEEE Aerosp. Electron. Syst. Mag., vol. 34, no. 6, pp. 16–27, 2019
  - B. He et al., "Use of the c-band microwave link to distinguish between rainy and dry periods," Adv. Meteor., vol. 2019, 2019
  - M. Xian et al., "Rainfall monitoring based on machine learning by earthspace link in the ku band," IEEE J. Sel. Topics Applied Earth Observ. Rem. Sens., vol. 13, pp. 3656–3668, 2020.
  - R. Giro et al., "Real-time rainfall estimation using satellite signals: Development and assessment of a new procedure," IEEE Trans. Instrum. Meas., vol. 71, pp. 1–10, 2022.
  - C. Gianoglio, et al., "Rain discrimination with machine learning classifiers for opportunistic rain detection system using satellite micro-wave links," Sensors, vol. 23, no. 3, p. 1202, 2023.

Indeed, the methods used for rain event detection for satellite microwave links are similar to many approaches used for CMLs and therefore relevant to this study. We agree to include this information and some examples from the literature list above in the introduction.

- Sec. 3.3: Performance indicators
  - is it really beneficial to introduce all those indicators? In principle, sensitivity (i.e. TPR according to the authors) and specificity (not considered by the authors) should be almost all we need. MCC can be useful as it is a global indicator combining FP and FN rejection, but it is not as straightforward as the previous two. It's not easy to state how good is an MCC value equal to 0.60-0.62 and how much an increase of MCC by 0.09 and 0.13 (I took these numbers from the abstract) is indeed valuable. Indeed, it is not obvious to make ratings of methods based on the MCC values in Figs. 5 and 7. Finally, please note that the importance or having high sensitivity rather than high specificity methods or vice-versa, depends on how wet/dry classification is used within the CML processing chain. Is it used to calculate the baseline? In this case, not misclassifying wet slots as dry is critical, i.e. sensitivity is the key indicator. I think the way you look at wet/dry classification as a part of data processing will drive the choice of the most significant performance indicator. Some discussion and a better justification of the indicators used is needed, instead of listing formulas and writing definitions.

The MCC is used because it is the best choice of a single score (rather than two or more) that acknowledges the imbalance between wet and dry events when considering the values of the confusion matrix. In the revised manuscript we will keep the MCC as the main score to compare the performance of different wet-dry detection methods and additionally add the false-positive rate (FPR) which complements the TPR. The pair of TPR and FPR is commonly known as the receiver operating characteristic (ROC). This will help readers unfamiliar with the MCC to interpret our results. We will drop the PPV, since, as the reviewer pointed out, not all scores are necessary and we will not add specificity as it can be directly taken from 1-FPR which will be included in the updated Figure 3. We will improve the justification of the used scores in the revised manuscript.

The choice of high sensitivity vs. high specificity is indeed depending on the application of the CML rainfall estimates which was discussed previously in Polz et al. 2020:

"All rain event detection methods have to make a similar trade-off: a liberal detection of wet periods is more likely to recognize even small rain rates, while it will produce more false alarms during dry periods. On the other hand, a conservative detection will accurately classify dry periods but is more likely to miss small rain events. One can address this by two means: by increasing detection rates on both wet and dry periods as much as possible and therefore decreasing the impact of the trade-off and by allowing the flexibility to easily adjust the model towards liberal or conservative detection, e.g., by only changing a single parameter."

In this study we aim to address the problem by "increasing detection rates on both wet and dry periods as much as possible and therefore decreasing the impact of the trade-off." The score of our choice to achieve this is the MCC which was also used to optimize the CNN method from Polz et al. 2020. We will add this explanation in the revised version of the manuscript.

  - ACC is shown in Fig. 3 (purple bars, I see small differences among bars) and in Fig. 9, first column. PCC is shown in Fig. 3 and Fig. 9 as well. Unless these

two highlight different aspects of the confusion matrix than MCC, I think it's simpler and better to show only MCC throughout.

We have used the PCC to compare rain rates derived from CMLs to the reference (RADKLIM-YW), hence we need to keep it. The ACC score will be removed from Figures 3 and 9 and will be shown only in the revised Figure 4.

- ○ The PPV indicator sounds a bit ambiguous to me in the framework of CML data, because it gets low (i.e. poor performance) if either there are a lot of FPs with respect to TPs, but it is also low for a given number of FPs if there are few TPs on the whole. FPs are often produced by factors other than rain, hence they have not in a strict relation with the number of TPs.

We agree and will remove the PPV score.

- ○ An interesting investigation would be to assess where errors (FP and FN occurrences) actually are. Are they at the start of an event rather than at the end? Or there are sequences of FPs far from events? Have the authors done such kind of analysis?

While an event-based analysis is indeed interesting, a detailed analysis is not straightforward due to some ambiguity in the definition of an event itself which is also depending on the resolution of the datasets in use. Currently, we believe that a larger analysis is necessary to treat this topic with enough care and it would certainly exceed the scope of the current manuscript which already treats a number of aspects related to the detection of rain events, such as daytime and rain rate dependencies.

- Figures 3,4,5,7: it's not a good idea to put "night" or "day" as labels on the y-axes of those figures. You should put the quantity displayed in the graph instead. Night and day should be placed into a text box in a free space over the figure or as descriptors after subfigure identifiers (e.g. a) , b), etc.) above each graph. Same for the x-axis: for instance, in Fig.3, the authors used different labels for the x-axis of subfigures a) and b): actually, they are the same axes. And they put TPR (y-axis) on the x- axis. The figures are indeed complex and they aggregate too much information spread in too many dimensions (night/day, percentage of rain occurrence as from SEVIRI products, rain intensity class, statistical indicator). Moreover, those small pictures on the top right of Fig. 3 are not well described: I guess here the TPR is weighted with the accumulated precipitation, that's why it is different from the blue histogram of Fig. 3b. Not straightforward really. It definitely looks too much and too difficult to track. Please simplify, dropping less meaningful dimensions. For instance, in Fig. 3a, TPR shows minor differences between night and day, as the authors state on page 10. Hence, one of the graphs of Fig 3a) can be dropped in my view. Also, I can't see any significant difference or trend between the two different SEVIRI products in Fig 3a.

We agree with the reviewer that Figures 3,4,5,7 are not easy to understand. We suggest remaking these figures based on these and the comments on the individual figures below. We suggest the following changes to all figures:

- Merge the day and night versions of one figure into one plot in Figures 3, 5, and 7 and omit the day/night split for Figures 4, 8, and 9

- Define and label each x and y axis correctly using the actual quantity that is displayed
- Reduce the visual scatter and provide easier-to-read figures
- Improve figure captions by giving more explicit descriptions

More changes are suggested in the answers to the comments on each figure below.

- Figure: 3: the authors show TPR of SEVIRI-based wet/dry classifier assuming radar-based classifier is the truth.
  - Is this comparison carried out mapping radar and SEVIRI pixels onto CML paths or it is just that the SEVIRI grid has been mapped onto the radar grid?

Both radar and SEVIRI grid have been averaged along each individual CML path weighted by the length of intersecting path segments in each pixel. This information was given in L168 but we suggest adding this information also in the figure caption and explaining it in more detail in the methods section. The code used for this path averaging is based on the pycomlink example which has additional illustrations:
https://github.com/pycomlink/pycomlink/blob/master/notebooks/Get%20radar%20rainfall%20along%20CML%20paths.ipynb

  - I didn't get the trend of TPR as a function of pNN, where NN is the probability of precipitation in percent (as the authors stated on page. 5), being pNN a SEVIRI product. So I expect TPR to increase with pNN, as SEVIRI wet/dry time series with high pNN have less wet slots than the ones with small pNN, hence a lower probability to incur in FNs. Why is it the opposite? On the other hand, the trend of PPV with pNN in Fig. 3b looks to agree with the feeling that a SEVIRI sample with high pNN is really a wet sample.

With higher NN fewer actual rain events are detected by SEVIRI which results in more FN and thus reduces TPR (for the same reason FPR is declining). Additionally, we suggest removing 3a) based on the general reviewer comment on Figures 3,4,5,7.

- Fig. 4: I have several comments here.
  - on Page 11, Line: 257 the author state: "The distribution of rain intensities and total precipitation amount of eight rain event detection methods is shown in Fig.4". It's way too generic. The explanation in the figure caption does not really help either (it should be improved as well). Finally, the word "count" in the x-axis label of the figure has nothing to do with rain intensities (following the logic of Fig. 3, count is the quantity displayed on the y- axis). So, what did you actually plot in Fig. 4? Is it a comparison between rain intensity/depth estimates across CML paths done by CML (2 methods)/SEVIRI (6 products) vs radar measurements? Or is it the percent difference between CML/SEVIRI wet counts and radar wet counts averaged over the population of a certain intensity class further divided by day and night? By the way, when you described SEVIRI data in Sec. 2.3, you didn't state clearly if precipitation intensity/depth are among SEVIRI products or retrievable from SEVIRI products., so I guess you are talking about counts. Please clarify and correspondingly edit your manuscript and it is really hard to get out of this based on what is written in explanations. Sec. 4.2.1 and in the figure caption.

In a revised version of the manuscript, we would remove Figure 4 as it is now because it is hard to understand. We think that the quality of binary rain event detection methods can be sufficiently explained by the updated versions of Figure 3 (comparing SEVIRI binary labels against the radar reference) and Figure. 5 (comparing the best SEVIRI products to the two TSB methods). Instead, we suggest a new figure showing ACC and relative bias for all individual ADB, TSB, and combined methods and the different intensity classes. We propose to add this figure to the last part of the results section. The new figure will contain information such as "Method XY detected X% of all moderate intensity events and the estimated CML rainfall in this class using method XY has a relative bias of X% compared to the radar reference".

   ○ In their explanation of the figure on page 11, the author state that "The SEVIRI-based ADB data sets behaved very similarly to the two TSB data sets RS and CNN". When I see it, TSB data work better than most of SEVIRI products except for the high-intensity class.

This will be changed with the suggestions from the previous comment about removing Fig 4. We also want to highlight that the new methodology will change part of the conclusion made here (i.e. stand-alone ADB performs worse than stand-alone TSB). More information on this change is summarized at the top of the answer to the reviewers comments.

   ● "a tendency of radar data to underestimate heavy rain intensities" (page 11 to justify CML/SEVIRI overestimate at large intensity) is a dangerous statement in my view. So far, the authors considered radar data as the truth and now they state that they maybe not good in estimating heavy rainfall. The author should provide a convincing evidence based on literature focused on German radar data, rather than citing a paper (Schleiss 2020) that worked over data from other radar networks.

We suggest weakening this statement and add more information e.g. on the used dataset giving more perspective on it. Also, we would add the consideration of the two cases for which we use radar data as a reference independently. We would communicate that the binary information from the radar is more trustworthy in our eyes than the exact rainfall estimate.

Results: in Fig. 9., the authors assessed the effectivity of a classifier based on mixing together TSB and ADB methods by performance indicators derived from the confusion matrix. The authors stated in the abstract "The separation of the attenuation time series in rainy and dry periods (rain event detection) is the most important step in this processing and largely determines the quality of the resulting rainfall estimates." So, a reliable classifier will end up in improving CML-based rainfall estimates. This is not demonstrated, however. It would be really good to see how Pc10all decreases the error on rainfall intensity estimates with respect to Pc10 and CNN through a scatterplot of rainfall intensity/depth as the ones in Fig. 9.

With the combination of ADB and TSB methods the MCC improves, PCC does not change strongly, but the relative bias changes (not shown in the previous version of the manuscript). Therefore we suggest adding the relative bias already to the new version of

Figure 4 to quantify the impact of ADB, TSB, and combined methods on the resulting rainfall estimates. This figure would be added to the last subsection of the results part.

We additionally propose to weaken the statement in the abstract in such a way that it emphasizes only that the chosen rain event method has a significant impact on the quality of the rainfall estimate instead of stating that it largely determines the quality.

**Technical corrections**

We completely agree with all technical corrections, except the ones which we commented on.

- Page 1, pp. 38-39: "Nevertheless, gauge-adjusted radar data is considered to be the best possible data basis", quite a strong statement if you ask me. I would be happy to state that RG calibrated radar data are considered reliable for estimating precipitation over large areas.

We agree to rephrase and weaken this statement, albeit we still think there is no better product we could use for this kind of analysis.

- Page 3, line 65: a full stop is missing
- Page 3, line 72: NWC SAF acronym not explained and written as one single word on following line 84.
- Page 3, line 74: "has carried out analyzes" is misspelled
- Page 5, line 125: "in Thoss" rather than "at Thoss"
- Page 5 line 126: a full stop is missing
- Page 7, line 189: "In step 1, we choose a method (either CML-TSB or SEVIRI-ADB) that shows a good average performance as a first guess"
- Page 7, line 193: Section rather than Chapter
- Page 10, Line 230: "To assess the quality of SEVIRI data it was 230 directly compared to radar data", this statement sounds a bit awkward: a comma is missing at least, or state it better.
- Page 10, Line 237: "over different rain intensity", would state "over different rain intensity values"
- Page 10, Lines 232-33: "TPR shows the percentage of wet RADKLIM-YW time steps per intensity class and precipitation amount, represented by SEVIRI data for different thresholds". Badly written. I am afraid this has to do with the comment I did above about the complexity of figures. I would state it in a simpler way: "Fig. 3a shows the TPR (in percent) of SEVIRI wet/dry classification at day (top) and night (bottom) divided per intensity class and per probability precipitation".
- Page 10, Line 240, "P30 showed the opposite". Cannot get it. p30 just shows that the p30 population of SEVIRI wet slot is closer to the population of RADAR wet slots then the p01 population (as PPV_p30> PPV_p01).

With this statement, we meant that while TPR is decreasing with a higher rainfall probability threshold, PPV is increasing. We suggested removing PPV based on a reviewer comment above and hence, would reformulate this statement using TPR and FPR, to discuss the impact of a chosen rainfall probability threshold.

- Page 13, lines 290-292: I guess dew effect is temperature dependent. I am not asking the authors to do such an analysis, but maybe processing data based on local temperature classes would help to clarify this point.

Indeed, dew formation depends on the temperature and dew point temperature. We found this phenomenon in many time series and some more information can be found in Polz et al. (2023). We think that analyzing this phenomenon would be out of scope for a revised version of the manuscript, but we can add the citation.

- Page 17, caption of Fig.8: "The rain intensities for FP and TP are estimated by the CML, while the rain intensities for FN are taken from the reference". I cannot get it. Why didn't you take rainfall intensities from the reference (i.e. radar) all the say? Please clarify.

There are no FP rainfall intensities from the radar reference. We will emphasize this in the revised figure caption.

Review of manuscript "Improved rain event detection in Commercial Microwave Link time series via combination with MSG SEVIRI data" by Andreas Wagner, Christian Chwala, Maximilian Graf, Julius Polz, Llorenç Lliso, José Alberto Lahuerta, and Harald Kunstmann.

OVERALL ASSESSMENT

I've read your well-written manuscript on improved rain event detection in commercial microwave link (CML) data with interest. The topic is highly relevant to improve wet-dry classification, and of the most important steps in CML rainfall retrieval. Especially for the Global South, where CMLs have the largest potential for improving rainfall information. Although some earlier work employs geostationary satellite data for rain event detection for CML data, this study is based on a much larger dataset with much wider coverage (3901 CMLs in Germany over a 4 month period). Moreover, it not only evaluates the performance of satellite data as an auxiliary-data-based (ADB) method, but also compares it with the performance of time-series-based (TSB) methods. Finally, also the combined use of the ADB and TSB method is evaluated employing a new method making use of liberal and conservative detection depending on the threshold. To conclude, this manuscript is a useful and innovative contribution to the field of CML rainfall estimation. I find it quite surprising that the satellite-based ADB methods have similar performance as TSB methods, especially because of representativeness errors (differences in sampling volume, parallax, et cetera) and inaccuracies in satellite precipitation probabilities. Below, I provide some suggestions, corrections and recommendations.

SCOPE, APPLICABILITY AND OUTLOOK

The combined use of a satellite-based ADB method and a TSB method gives (slightly) better results compared to TSB methods only. The analyses are based on data from Germany. Below a couple of thoughts that could be incorporated to better frame the manuscript in the introduction or that could lead to additional recommendations:

1. This study is of course relevant for CML rainfall estimation in countries where TSB methods based on high temporal sampling can be applied.

We suggest mentioning this helpful comment in the conclusion.

2. Now the manuscript seems mostly relevant for Germany, but Germany could also be seen as a testbed with relatively good gauge-adjusted radar reference data. Hence, this study is especially relevant for the Global South, where usually no other auxiliary (near real-time) data exist (ground-based weather radars and rain gauge data are sparse). In the Global South, part of the CML network is in rural areas, where densities may be too low to apply a "nearby-link" approach. In addition, often 15 min data are available, which limits the applicability of TSB methods. Because the performance of satellite-based rain event detection is evaluated and shown to be useful, this manuscript is especially relevant for the Global South.

Indeed, this was actually one of our main motivations and we should state this more clearly in the manuscript. Therefore, we suggest adding this in the revised discussion of the ADB results as well as in the motivation of the objectives in the introduction and the conclusion.

3. One could recommend to test a combined satellite-based ASB method and compare it to the "nearby-link" approach, and also to combine both methods and evaluate its performance. This would especially be relevant for the Global South.

We agree with the relevance but suggest not adding the nearby-link approach to this study. First, a suitable CML dataset with 15-minute min-max data has to be found. The usage is 1-minute instantaneous data or simulated 15-minute min-max data from 1 minute data, as we have available in Germany, with the nearby approach could be a study on its own. Second, a further comparison of additional methods for rain event detection would exceed the scope of this manuscript. We will add this idea as potential future work in the outlook of the conclusion.

4. The authors state that "best results are usually achieved with radar data". These data are available for Germany. One could recommend to explore the use of radar data for rain event detection and also combine it with a TSB method. Naturally, when no radar data are employed, the results for CML rainfall estimation in Germany solely show the performance of CML, which is relevant (testbed), and this already provides good rainfall estimates. So this manuscript is not only relevant for other regions, but also for Germany as such. Results for Germany could improve, though, when radar data would be employed for wet-dry classification.

Indeed, using radar data as a rain event classification could be an interesting option and we are not aware of a published evaluation on this. Nevertheless, the main focus of this study is the performance of the SEVIRI products and their combinations with TSB methods. We propose to do some initial tests and, depending on the feasibility and added benefit for the current study we will decide whether to add metrics to figures where rainfall rates are analysed.

5. What do the authors expect in terms of performance of satellite-based ADB methods when data from the new Meteosat Third Generation (https://www-cdn.eumetsat.int/files/2020-04/pdf_mtg_info-pack_v12.pdf) would be employed? Its higher spatial and temporal resolution is beneficial and closer to the spatiotemporal resolution of CMLs, although parallax will still give rise to representativeness errors, especially in the mid-latitudes. Perhaps that more channels will allow for improvement in precipitation probability estimation. It could at least be worth mentioning MTG around L. 384.

We suggest adding information on the new product and expected outcomes in the conclusion as suggested by the reviewer. The general prospect is a higher quality of the ADB methods, especially for rainfall events that exhibit smaller spatial and temporal scales.

METHOD AND RESULTS

1. Why did the authors choose these MSG products? The use of a geostationary product is clear because of its 15 min, or better, temporal resolution. But other products exist.

Other precipitation products indeed exist, but we wanted to use a product with precipitation probability instead of a precipitation product with rain rates for which we would have had to add a threshold at a low rain rate (at which SEVIRI rainfall products have limited accuracy) to distinguish between wet and dry. PC and PCPH from MSG SEVIRI are the only

precipitation probability products available for our region of interest. Compared to a pure precipitation product, the precipitation probability products enabled us to consistently alter the classification threshold similar to how it can be done for certain TSB methods. We suggest adding this reasoning in the data section of the revised manuscript.

2. L. 145: "This baseline is the last dry time-step of the TRSL time-series" suggest that the baseline is based on only 1 data point. In not, please clarify which period preceding a rainfall event is considered for the computations, and how many dry time steps / data points are needed for its computation.

Indeed we used one data point for the baseline estimation. This was also done e.g. by Graf et al. 2020. We tested the baseline approach from the nearby approach (median of all dry time steps over the last 24 hours) and found that it did not work well with 1-minute instantaneous data. Possible reasons could be the greater amount of fluctuations in the data with higher resolution and that the nearby approach typically uses the mean between minimum and maximum power.

3. L. 154: Good agreements with what? Clarify, e.g., the CML rainfall estimates with ITU parameters compare well with reference data.

We suggest adding the reference data (RADOLAN RW, a gauge-adjusted radar product) used in the mentioned studies.

4. L. 159: Standard deviation of what variable? TSL, RSL, TRSL?

We suggest adding TRSL (TL as defined by Fencl et al. 2023) to this sentence to clarify it.

5. L. 233: "the largest differences": add "between the chosen probabilities".

We suggest to be more precise in the description of the results of a revised version of Fig. 3.

6. L. 263: "behaved very similarly" seems especially the case for most graphs for p01, but larger differences are found for p10 and p20.

Similar to the previous comment we also suggest updating the description here. This will change a bit as we will use the binary wet/dry information directly and not the one derived from rain rates as explained in the general comments at the top of the answers to the reviewers..

7. General remark: the readability of figures would be improved if the variable name and unit would be added to the vertical axes (Figures 3, 4, 5, 7).

We agree that these Figures can be improved and summarized changes here (see also the respective comments of reviewer #1):

- merge day and night rows into one plot
- define and label each x and y axis correctly
- reducing the visual scatter and providing easier-to-read figures
- improve figure captions by giving more explicit descriptions

8. You could consider using the present tense instead of the past tense when describing results.

We think that the results that have been derived in the past should be described in the past tense. Generally applicable statements can be put in the present tense.

9. L. 296-297: rephrase a bit, because the ADB method itself is completely independent from the CML time series, but it is about the effect of CML time series after applying a rain event detection method.

As we suggest analyzing the binary wet/dry labels and not rain rates >0.1mm/h as wet (as we stated in our general statement on the top of the answers the reviewer comments), this sentence should be rephrased in a revised manuscript.

10. Caption Figure 6: make clear that the green line is a CML-based rainfall intensity.
We will revise this figure more substantially to make it easier to understand.

11. L. 320-322: I find these lines difficult to follow. Can you explain more clearly how PC10all is obtained?

We suggest explaining the combination of methods in a more concise way and will update Figure 6 to complement Figure 2 and make it more easy to understand.

12. L. 325: it seems that one selection is made from the best of six combinations. Do the "three data sets" represent TSB only (CNN), ADB only (PC10), and the TSB & ASB combination (PC10all)?

Yes, this was true. We suggest stating this more explicitly in a revised version of the manuscript.

13. L. 332: it seems at most ~0.15, and especially an improvement with respect to RS and a slight improvement with respect to CNN.

As the results of the combined methods will change slightly with the suggested changes in the general statement about necessary corrections of the methods, we will adjust the quantitative statements accordingly.

14. Caption Figure 7: "Same as Fig. 4" should be "Same as Fig. 5".

As already stated above, we suggest making the captions more readable and explicit.

15. Figure 9: make square plots, so that the horizontal and vertical axes get the same scale and the dashed x-y line gets an angle of 45 degrees. Then it becomes more obvious that pc10all outperforms cnn. It would facilitate the comparison of performance between methods.

Agreed.

16. The authors could provide mean and/or median values of the metrics in Figure 9 for both datasets in each figure, to summarize the performance of each method. And if I understand correctly, this is already presented in Figure 7 for MCC. So, the reader could be pointed again to Figure 7 to emphasize that it contains the MCC metric for CNN, PC10 and PC10all (which are shown in Figure 8) and to help the reader to find the overall results.

We will improve this figure by adding a point that represents the mean of the metric from both datasets. Figure 7 shows the score computed for the whole dataset which may differ from the average of individual CML scores as shown in the revised Fig. 9 (Fig. AC2).

17. I probably missed it, but what is the time interval or duration for which the rain event detection is evaluated? Is it 15 min or 1 h? When comparing to radar data, it is beneficial to use a somewhat longer duration, such as 60 minutes, to limit representativeness errors (e.g., differences in sampling volume, time it takes for precipitation to fall from the radar sampling volume to the Earth's surface, advection of precipitation). This is at least common for rain gauge versus radar comparisons. In the case of CMLs, one could argue that its scale is somewhere between the point and radar grid cell size, making it less vulnerable to representativeness errors.

All results are shown for a 15-minute time scale, as this is the temporal resolution available for SEVIRI. We believe that the 15-minute resampling of the radar data as well as the fact that it is adjusted to rain gauges helps to mitigate the issues mentioned by the reviewer. Longer aggregations, e.g. to 1 hour as suggested by the reviewer, would certainly mitigate the issues even more. However, for the validation of rain event detection in CML data, one has to find a balance between keeping a high temporal precision while not being too inaccurate. We work a lot with the RADKLIM-YW data at its native 5-minute resolution and typically find that temporal offsets of some minutes can occur in certain situations (high measuring height of radar, potentially combined with strong advection) when comparing with CML data. For potential future real-time applications, low-latencies, i.e. short time windows for rain event detection are also a requirement.  Therefore we would keep the time scale of the evaluation at 15 minutes.
We suggest adding a description and justification of the time interval used for evaluation to the method section.

REMAINING MINOR REMARKS:

- L. 28: A tipping bucket represent one important and frequently used type of rain gauge, but these are probably not considered the best rain gauges given limitations in sampling of low- and high-intensity rainfall related to the tipping bucket volume and the number of possible tips. So stating that these tipping buckets provide "usually provide the best point measurements of precipitation" will not generally be true. This statement could be weakened a bit.

We agree to weaken this statement.

- For instance, at L. 44 "precipitation intensity" is used, whereas CMLs are typically useful for estimating "rainfall intensity". In case "precipitation intensity" is kept, I suggest to mention

once in the introduction that CMLs and the employed algorithm are typically suited for rainfall estimation, i.e., liquid precipitation estimation.

We suggest mentioning the limitations of CMLs and also replacing precipitation with rainfall throughout the manuscript.

- L. 116: "No specific device for precipitation measurement is on-board": could be made more specific (space-borne radar and/or radiometer).

We agree and suggest specifying this statement.

- L. 147: "radome" is typically used when referring to the protective cover of weather radars. Perhaps "cover" is a more appropriate word for CML antennas.

We suggest rephrasing "radome" to "cover".

- Could the authors add some information on the availability of the radar, CML and satellite data sources (e.g., expressed as a percentage)?

We suggest adding this information to a table that gives information on the temporal resolution, pixel size, etc. on all used data sources as requested by the other referee.

- Data availability: are the geostationairy satellite data publicly available? If so, please add where the data can be retrieved.
The PC and PCPH products are provided by the co-authors from Agencia Estatal De Meteorología who are part of NWC SAF. Recent data is freely available, but long-term records must be requested individually. We suggest adding this information to the data availability section of the manuscript.

- Titles in the reference list: the use of capital letters is not consistent, e.g., Steiner et al. (2004) versus Wang et al. (2012).

We suggest to revise the reference list accordingly.

References

- Fencl, M., Nebuloni, R., CM Andersson, J., Bares, V., Blettner, N., Cazzaniga, G., ... & Zheng, X. (2023). Data formats and standards for opportunistic rainfall sensors. *Open Research Europe*, *3*, 169.
- Graf, M., Chwala, C., Polz, J., & Kunstmann, H. (2020). Rainfall estimation from a German-wide commercial microwave link network: optimized processing and validation for 1 year of data. *Hydrology and Earth System Sciences*, *24*(6), 2931-2950.
- Leijnse, H., Uijlenhoet, R., & Stricker, J. N. M. (2008). Microwave link rainfall estimation: Effects of link length and frequency, temporal sampling, power resolution, and wet antenna attenuation. *Advances in Water Resources*, *31*(11), 1481-1493.

- Polz, J., Chwala, C., Graf, M., & Kunstmann, H. (2020). Rain event detection in commercial microwave link attenuation data using convolutional neural networks.

---

## Referee Report (RR1)

REVIEW OF THE PAPER "Improved rain event detection in Commercial Microwave Link time series via combination with MSG SEVIRI data", AMT 2023-175 (round II)

**General comment**

The authors did a huge effort in editing the manuscript according to reviewers comments. I think that this new version has improved a lot and it is much clearer than the original one. I also would like to thank the authors for their detailed replies to my questions. There are still a few points, which, in my opinion, are due a **minor revision**.

- Sampling and resampling of SEVIRI and CML data: for wet /dry classification (ADB methods), SEVIRI has been resampled to 1-min, i.e. the same as CML data. However, comparison against radar data is carried out resorting CML to 15-min sampling. Why you did not use 15-min all the way, just resampling CML data?

- About my comment on performance indicators in the first round of review. Specifically about MCC. It is not a matter of being familiar or not with it. I think that several readers would not be able to rate the statement in the abstract "Compared to basic and advanced TSB methods, these combinations improved the Matthews Correlation Coefficient of the rain event detection from 0.49 (0.51 resp.) to 0.59 during the day and from 0.41 (0.50 resp.) to 0.55 during the night". Is it a significant/huge improvement or not? Are 0.49 or 0.41 acceptable values for the MCC? To help understanding MCC, if we compare (6) and (1), we get that MCC is basically PCC for binary data. That would be a synthetic and easy explanation of MCC. When I see it from the reader's perspective, it would be more effective to summarize in the abstract the improvement brought by combinations using TPR and FPR, or just writing a simple statement as the one on p. 19 lines 420-21.

- About my general comment on wet/dry rationales for radar data: the authors explain how radar pixels are combined (p. 5, line 125) to overlap a CML as the comparison with SEVIRI is done over CML paths. The radar-based precipitation value is derived by a weighted average in space (according to the fraction of the path overlapping the radar pixel) and an arithmetic average in time (from 1 to 15-min). However, it is not well explained how radar time series (i.e. rainfall rate estimates) were reduced to wet/dry time series for the validation of SEVIRI products in Fig. 4. It is stated that wet/dry threshold on radar data is 0.1 mm/h (p. 4 line 130). Hence, I guess the authors first calculated the radar-based rainfall estimate over the CML path and then they thresholded it at 0.1 mm/h. I think that this procedure should be explained in the text for instance on p.5 after line 125.

- Threshold on SEVIRI probability of rain. As the authors said in Sec. 5.1, it is surprising that PC and PC-Ph work at their best with such a low probability of rain. This looks even more surprising considering that it is calculated over a rather large pixel (table 1). Can the author provide any information about how these precipitation products were extracted from SEVIRI measurements to justify these outcomes?

- Figure 7: I cannot get why the relative bias of each class is normalized to the average rainfall intensity over all classes (denominator of Eqn. 2) if I got it, which I am not sure. If we assume an order of magnitude 1000 mm of rainfall per year, the average rainfall intensity would be around 0.12 mm/h. Is it correct? In my view, for class, say, light 2, the denominator should be the average of the occurrences of radar-based rainfall rates between 1 and 2.5 mm/h. The important information I retain from Fig. 3c is the sign of the relative bias. The height of bars is about the balance between negative and positive errors. In the evaluation of CML rain estimates time vs radar, using different wet/dry classifiers, I think a useful indicator is missing, that is the RMSE, which would help in assessing also the performance over individual classes.

- One thing I guess remains without an explanation is why all methods underestimate precipitation with respect to the radar reference whatever the rainfall intensity class, while they overestimate dry periods (relative bias in Fig.7). I think a comment in the manuscript is due. Even if this fact is explained somewhere else, please not only add only the reference, but at least one explanatory statement.

**Specific comments**

- P.4, line 144: "it is calculated by a regression of IR and Water Vapour channels (WV)." What you mean by WV channel? Channels are identified by a frequency.
- P. 7, line 171: "We computed RS and CNN on a 1-minute basis". Do you mean in previous paper?
- P. 7, line 175 what you mean by "we forward filled?" You just classified all minutes within a 15-min SEVIRI wet slot as wet? It could be a problem when it starts/stops raining or during intermittent rain. This point has to with the first bullet in my general comments.
- P. 10 Eqn. (2) I think the terms on the numerator should be switched, as all methods underestimate rain intensity as the authors state several times, it means that RB<0
- P. 12 lines 299-301 and P. 19 line 406-408: the authors do not bring a physical evidence that dew formation is the responsible for such a drop of the MCC for RS from day to night. They just say that the difference between RS and CNN performance suggests this conclusion. I suggest to smooth the statement on p. 19 which sounds like an harsh statement. (also I think the word "assumption" is not correct in this context)
- P. 13 Figure 4 caption: better to add that TPR, FPR, MCC refer to wet/dry classification performance while PCC is for rainfall intensity estimate.
  P. 13 line 316-319: from Figs. 5 and 7, as far as I see it, the three combined methods shown perform the same except for the bias. Moreover, the 10.8% bias is attributed to PC1 combined, while from Fig. 7 it seems the one of CNN-combined.

**Technical corrections**

- p. 6 line 163, "adapted"
- p. 7, line 185: I guess it is "Fig. 2"
- p. 10, line 219 "For comparison against the benchmark" sounds better
- P. 12, line 2929: "one" instead of "on"
- P. 14, line 326-27: I guess you are referring to Fig 7 c) and d)

---

## Author Response (AR2)

We would like to thank the reviewer for the second revision of our manuscript and the helpful comments that further improved the manuscript. We agree with most of the raised points and questions and clarified or rephrased the manuscript accordingly. We additionally changed the color scheme of Fig. 6 and 8 to be more comprehensible to people with color vision deficiencies.

In the following the reviewers' comments are in black, our answers in blue, and quotation marks additionally mark the text changes in the manuscript.

REVIEW OF THE PAPER "Improved rain event detection in Commercial Microwave Link timeseries via combination with MSG SEVIRI data", AMT 2023-175 (round II)

General comment

The authors did a huge effort in editing the manuscript according to reviewers comments. I think that this new version has improved a lot and it is much clearer than the original one. I also would like to thank the authors for their detailed replies to my questions. There are still a few points, which, in my opinion, are due a minor revision.

- Sampling and resampling of SEVIRI and CML data: for wet /dry classification (ADB methods), SEVIRI has been resampled to 1-min, i.e. the same as CML data. However, comparison against radar data is carried out resorting CML to 15-min sampling. Why you did not use 15-min all the way, just resampling CML data?

There are good reasons why CML data is processed at a 1-minute resolution and evaluated at a 15-minute resolution:
The processing at a 1-minute resolution is due to compatibility with established processing methods. The reference methods (CNN, q80), but also the wet antenna attenuation and baseline estimation were either developed, or calibrated, and tested for one-minute instantaneous CML data (Graf et al. 2020, Polz et al. 2020). They would have to be adapted to a lower resolution, which is not our aim here. We aimed to see if SEVIRI data, with its 15-minute resolution, can be used for rain event detection with the CML data we have available. It was straightforward to downsample the SEVIRI wet-dry indicator to a 1-minute resolution leaving the rest of the processing unaltered and we could show that this approach leads to quite satisfactory results.
To evaluate the binary wet-dry information from SEVIRI (with its 15-minute resolution) we relied on the radar reference which already has a lower temporal resolution (5-minute) than the CML data and, therefore, we decided to further resample to a 15-minute resolution. To improve the comprehensibility of the study we compared all results at the same resolution. To improve the clarity on this we added an explanation to the last section of 3.1.1 Individual methods for rain event detection:
"We forward-filled the 15-minute classification to a 1-minute resolution in the CML processing described in Sec. 3.2. This temporal resolution is necessary for the two TSB methods and other CML processing methods such as WAA compensation and baseline estimation as they were developed and tested for this resolution (Graf et al. 2020)."

- About my comment on performance indicators in the first round of review. Specifically about MCC. It is not a matter of being familiar or not with it. I think that several readers would not be able to rate the statement in the abstract "Compared to basic

and advanced TSB methods, these combinations improved the Matthews Correlation Coefficient of the rain event detection from 0.49 (0.51 resp.) to 0.59 during the day and from 0.41 (0.50 resp.) to 0.55 during the night". Is it a significant/huge improvement or not? Are 0.49 or 0.41 acceptable values for the MCC? To help understanding MCC, if we compare (6) and (1), we get that MCC is basically PCC for binary data. That would be a synthetic and easy explanation of MCC. When I see it from the reader's perspective, it would be more effective to summarize in the abstract the improvement brought by combinations using TPR and FPR, or just writing a simple statement as the one on p. 19 lines 420-21.

On the one hand, the actual value of MCC (as PCC or many other metrics) cannot easily be put into categories of good or bad because they depend on the use case. In some cases, a MCC of 0.8 might be really bad while for other use cases, a MCC of 0.2 might already show a success. Some rain event detection method studies already used the MCC, allowing for a comparison of the values. On the other hand, we agree that for a reader it is easier to understand a qualitative statement. Therefore, we added one sentence to the abstract with a qualitative explanation of the results of the manuscript:
"Additionally, these combinations increased the number of true positive classifications, especially for light rainfall compared to the TSB methods, and reduced the number of false negatives while only leading to a slight increase in false positive classifications."

- About my general comment on wet/dry rationales for radar data: the authors explain how radar pixels are combined (p. 5, line 125) to overlap a CML as the comparison with SEVIRI is done over CML paths. The radar-based precipitation value is derived by a weighted average in space (according to the fraction of the path overlapping the radar pixel) and an arithmetic average in time (from 1 to 15-min). However, it is not well explained how radar time series (i.e. rainfall rate estimates) were reduced to wet/dry time series for the validation of SEVIRI products in Fig. 4. It is stated that wet/dry threshold on radar data is 0.1 mm/h (p. 4 line 130). Hence, I guess the authors first calculated the radar-based rainfall estimate over the CML path and then they thresholded it at 0.1 mm/h. I think that this procedure should be explained in the text for instance on p.5 after line 125.

We agree with the reviewer and added a sentence explaining the application of the threshold at the suggested location:
"For the usage as a binary wet-dry reference, we used a rainfall intensity threshold of 0.1 mm/h at the 15-minute resolution. All values below 0.1 mm/h were considered dry."

- Threshold on SEVIRI probability of rain. As the authors said in Sec. 5.1, it is surprising that PC and PC-Ph work at their best with such a low probability of rain. This looks even more surprising considering that it is calculated over a rather large pixel (table 1). Can the author provide any information about how these precipitation products were extracted from SEVIRI measurements to justify these outcomes?

As the SEVIRI products are indirect rainfall indicators based on optical and infrared measurements an estimate is based on observed cloud mask, cloud optical thickness, cloud water path, cloud top temperature, etc. As a result, they have large uncertainties in detecting light rainfall and the exact outline of precipitation fields. However, for Europe the cloud mask product has a very high rate of detection of dry pixels, rejecting most dry areas for rainfall

anyway (leading to a small number of FP in the final rain event detection). For lower thresholds of PC and PC-Ph, the increase in TPR mostly equals the increase in FPR as shown in Fig. 4 with P10 as the best product.
"For the SEVIRI products, this does not seem to be the case. One reason might be that the cloud mask, which both products use, has good performance over Europe and therefore has a high probability of true dry detection, resulting in a low number of FPs in general and thus also for low thresholds."

- Figure 7: I cannot get why the relative bias of each class is normalized to the average rainfall intensity over all classes (denominator of Eqn. 2) if I got it, which I am not sure. If we assume an order of magnitude 1000 mm of rainfall per year, the average rainfall intensity would be around 0.12 mm/h. Is it correct? In my view, for class, say, light 2, the denominator should be the average of the occurrences of radar-based rainfall rates between 1 and 2.5 mm/h. The important information I retain from Fig. 3c is the sign of the relative bias. The height of bars is about the balance between negative and positive errors. In the evaluation of CML rain estimates time vs radar, using different wet/dry classifiers, I think a useful indicator is missing, that is the RMSE, which would help in assessing also the performance over individual classes.

About the relative bias and the normalization:
Eqn. 2 was slightly wrong because it should be total error divided by total precipitation (we corrected this now). We hope this resolves the understandable confusion.
The main purpose of Fig. 7 is to analyze the "Performance of rain event detection methods for different rain intensity classes". The relative bias shown in this figure is the mean error with respect to the radar reference in that class divided by the total radar rainfall intensity of all classes. This way, we can see how this overall bias for each method (Fig 7d) is a sum of the bias each method has in the individual classes. Therefore, we can also see where improvements are more important. If we normalized each class by the "average of the occurrences of radar-based rainfall rates between 1 and 2.5 mm/h" we would not gain additional knowledge, but we would lose an overview of where the overall bias comes from.
The updated Figure caption is:
"The relative bias in each class is the sum of all errors in one class as a percentage of the total rainfall of the full reference data. This way, the five leftmost classes add up to the 'all' class on the right."

About the use of RMSE:
The Figure below shows the draft of an updated Fig 7 that includes the RMSE.

[Figure]

In general, the only reason not to include this information is the added length of the manuscript and the resulting difficulty in extracting useful information from the large number of indicators presented. As you can see, an RMSE comparison between intensity classes is not possible because of the scaling of the RMSE with the rainfall intensity. Additionally, the different magnitudes of the RMSE in the different classes make a comparison of methods in one class harder. Therefore, we decided not to add the RMSE to the analysis.

- One thing I guess remains without an explanation is why all methods underestimate precipitation with respect to the radar reference whatever the rainfall intensity class, while they overestimate dry periods (relative bias in Fig.7). I think a comment in the manuscript is due. Even if this fact is explained somewhere else, please not only add only the reference, but at least one explanatory statement.

Thank you for pointing this out. There are good reasons for the observed biases in the different classes which we will explain in the following:

Let's first assume that derived rainfall rates during correctly detected rain events perfectly fit to the rainfall reference. Then the effect of a negative bias would still be visible because of false negatives since the rainfall intensity classes only include TP and FN periods. A part of this negative bias is compensated by FPs in the dry class where only a positive bias is possible due to the radar reference rainfall being 0. With this assumption laid aside, a bias for estimated rainfall rates is also present which may be due to WAA compensation. The two combined products have almost no overall bias compared to the reference. All others show a negative bias, that is true. This fact can potentially be attributed to WAA compensation. We therefore added a statement to Section 4.3:

"The contribution of the different intensity classes to the overall relative bias in Fig. 7 shows that the overestimation due to FPs in the dry class is smaller than the underestimation in the positive intensity classes where FNs are one major factor for missed rainfall. This way, all methods except PC01-combined show an overall negative bias. One additional explanation for an overall negative bias could be a too-large compensation for WAA. Tiede et al. (2023) described strong WAA fluctuations during long-lasting rain events, a fact that is not considered in the WAA compensation method we chose (or any other available method). Therefore, there is some uncertainty in WAA compensation which influences the bias."

Specific comments

- P.4, line 144: "it is calculated by a regression of IR and Water Vapour channels (WV)." What you mean by WV channel? Channels are identified by a frequency.

While this is technically true, the two IR channels at 6.2 and 7.3 μm from MSG SEVIRI are commonly called water vapor channels see e.g. https://www.nwcsaf.org/pc-ph_description or https://www-cdn.eumetsat.int/files/2020-04/pdf_conf_p46_s3_04_georgiev_v.pdf

- P. 7, line 171: "We computed RS and CNN on a 1-minute basis". Do you mean in previous paper?

We clarified this point by changing the sentence to:
"Identically to Polz et al. (2020), we computed RS and CNN based on 1-minute TL data."

- P. 7, line 175 what you mean by "we forward filled?" You just classified all minutes within a 15-min SEVIRI wet slot as wet? It could be a problem when it starts/stops raining or during intermittent rain. This point has to with the first bullet in my general comments.

We think we answered this point already below the first bullet point in the general comments.

- P. 10 Eqn. (2) I think the terms on the numerator should be switched, as all methods underestimate rain intensity as the authors state several times, it means that RB<0

Indeed, we switched r_ref and r_cml in the numerator.

- P. 12 lines 299-301 and P. 19 line 406-408: the authors do not bring a physical evidence that dew formation is the responsible for such a drop of the MCC for RS from day to night. They just say that the difference between RS and CNN performance suggests this conclusion. I suggest to smooth the statement on p. 19 which sounds like an harsh statement. (also I think the word "assumption" is not correct in this context)

We have worked extensively with CML data and have often seen dew formation and the described behavior of the RS and CNN rain event detection methods. We have also investigated these events using temperature, humidity, and dew point temperature from ERA5 (Polz et al. 2023). While the RS often classifies dew events as rainy because it cannot distinguish between high-frequency (rain-induced attenuation) and low-frequency changes of the signal (caused by dew), CNN was able to learn that they are not. However, we will weaken the statement in the manuscript, as this is not a very important point to the overall conclusions of the manuscript. We rephrased the section to:
"While CNN was able to perform equally well during nighttime, the other TSB method RS showed a decreased performance. One possible explanation could be the formation of dew on the antennas during nighttime that can regularly be observed in CML time series as a slowly increasing (after sunset) and decreasing (after sunrise) attenuation. The more sophisticated pattern recognition algorithm of the CNN method seems to be able to correctly classify these periods as dry."

- P. 13 Figure 4 caption: better to add that TPR, FPR, MCC refer to wet/dry classification performance while PCC is for rainfall intensity estimate.

Good idea, we rephrased this caption to:

"Performance metrics of the binary rain event classification (TPR, FPR, and MCC) and the rainfall rates (PCC) of the PC (blue) and PC-Ph (orange) products compared to the radar reference. The results of each score are presented for different thresholds (x-axes) and split into day (light colors) and night (dark colors)."

- P. 13 line 316-319: from Figs. 5 and 7, as far as I see it, the three combined methods shown perform the same except for the bias. Moreover, the 10.8% bias is attributed to PC1 combined, while from Fig. 7 it seems the one of CNN-combined.

For the MCC (Fig 5) there are only small differences, but the accuracy and bias for different rainfall intensities (Fig 7) show larger differences, we therefore kept the three shown combinations. Regarding the description of the rel. bias in Fig. 7 we accidently mixed the rel. bias values for PC01-combined and CNN-combined up. We now placed the correct value and product together:
"PC10-combined also showed the lowest relative bias of -2.1% compared to PC01 combined with 2.6% and CNN combined with -10.8% (see Fig. 4d)."

Technical corrections
- p. 6 line 163, "adapted"

Corrected this mistake.
- p. 7, line 185: I guess it is "Fig. 2"

Corrected this mistake.
- p. 10, line 219 "For comparison against the benchmark" sounds better

Since we have not used the word "benchmark" anywhere else in the manuscript, we do not want to use it here.
- P. 12, line 2929: "one" instead of "on"

Corrected this mistake.
- P. 14, line 326-27: I guess you are referring to Fig 7 c) and d)

Corrected this mistake.

**References**
Graf, M., Chwala, C., Polz, J., & Kunstmann, H. (2020). Rainfall estimation from a German-wide commercial microwave link network: Optimized processing and validation for 1 year of data. *Hydrology and Earth System Sciences*, *24*(6), 2931–2950. https://doi.org/10.5194/hess-24-2931-2020

Polz, J., Chwala, C., Graf, M., & Kunstmann, H. (2020). Rain event detection in commercial microwave link attenuation data using convolutional neural networks. *Atmospheric Measurement Techniques*, *13*(7), 3835–3853. https://doi.org/10.5194/amt-13-3835-2020

Polz, J., Glawion, L., Graf, M., Blettner, N., Lasota, E., Schmidt, L., Kunstmann, H., & Chwala, C. (2023). Expert Flagging of Commercial Microwave Link Signal Anomalies: Effect on Rainfall Estimation and Ambiguity of Flagging. *2023 IEEE International Conference on Acoustics, Speech, and Signal Processing Workshops (ICASSPW)*, 1–5. https://doi.org/10.1109/ICASSPW59220.2023.10193654

Tiede, J., Chwala, C., & Siart, U. (2023). New Insights Into the Dynamics of Wet Antenna Attenuation Based on In Situ Estimations Provided by the Dedicated Field Experiment

ATTRRA2. *IEEE Geoscience and Remote Sensing Letters, 20*, 1–5. https://doi.org/10.1109/LGRS.2023.3320755